# Diverse modes of H3K36me3-guided nucleosomal deacetylation by Rpd3S

Haipeng Guan[1,2,9], Pei Wang[1,2,9], Pei Zhang[1,2], Chun Ruan[3], Yutian Ou[1,2], Bo Peng[1,2], Xiangdong Zheng[4], Jianlin Lei[2,5], Bing Li[6 ✉], Chuangye Yan[2,7,8 ✉] & Haitao Li[1,2,8 ✉]

Context-dependent dynamic histone modifications constitute a key epigenetic mechanism in gene regulation[1–4]. The Rpd3 small (Rpd3S) complex recognizes histone H3 trimethylation on lysine 36 (H3K36me3) and deacetylates histones H3 and H4 at multiple sites across transcribed regions[5–7]. Here we solved the cryo-electron microscopy structures of *Saccharomyces cerevisiae* Rpd3S in its free and H3K36me3 nucleosome-bound states. We demonstrated a unique architecture of Rpd3S, in which two copies of Eaf3–Rco1 heterodimers are asymmetrically assembled with Rpd3 and Sin3 to form a catalytic core complex. Multivalent recognition of two H3K36me3 marks, nucleosomal DNA and linker DNAs by Eaf3, Sin3 and Rco1 positions the catalytic centre of Rpd3 next to the histone H4 N-terminal tail for deacetylation. In an alternative catalytic mode, combinatorial readout of unmethylated histone H3 lysine 4 and H3K36me3 by Rco1 and Eaf3 directs histone H3-specific deacetylation except for the registered histone H3 acetylated lysine 9. Collectively, our work illustrates dynamic and diverse modes of multivalent nucleosomal engagement and methylation-guided deacetylation by Rpd3S, highlighting the exquisite complexity of epigenetic regulation with delicately designed multi-subunit enzymatic machineries in transcription and beyond.

Dynamic histone modifications and their crosstalk have critical roles in gene regulation[1–4]. For example, histone H3 lysine 4 trimethylation (H3K4me3) and H3K36me3 are hallmarks that demarcate transcriptional initiation sites and coding regions across a transcription unit. The histone methyltransferase Set2 can be directly recruited by elongating RNA polymerase II and specifically methylates H3K36[8–10]. The Rpd3S histone deacetylase (HDAC) complex recognizes H3K36me3 at coding regions and suppresses transcription from cryptic promoters via its deacetylase activity in yeast[5–7]. Thus, the Set2–Rpd3S regulatory axis establishes a crosstalk between H3K36 methylation and histone deacetylation to inhibit abnormal transcription[11–13].

Rpd3, which was originally identified as a global gene regulator and a co-repressor[14], was first reported to function as a HDAC in 1996 (ref. 15). This discovery, along with the characterization of Gcn5 as the first histone acetyltransferase linked to transcription, established a new paradigm of chromatin biology and epigenetics. Yeast Rpd3, a class-I HDAC family member, could assemble with Sin3, me1, Rco1 and Eaf3 to form a 0.6-MDa Rpd3S complex[7,16]. Meanwhile, Rpd3, Sin3 and Ume1 can also assemble with a different set of factors, including Pho23, Cit6 and Ume6, to form a 1.2-MDa Rpd3 large (Rpd3L) complex in yeast. Both Rpd3S and Rpd3L contribute to gene repression, depending on the methylation context of H3K36me3 versus H3K4me3[17]. The Rpd3S complex recognizes H3K36me3 through the chromodomain (CHD) of the Eaf3 subunit (CHD$_{Eaf3}$), as well as the unmodified histone H3 N-terminal tail through the plant homeobox domain (PHD) finger of Rco1 across coding regions[18,19]. Histone H3K36me3 promotes the deacetylase activity of the Rpd3S complex towards histones H3 and H4, which is associated with coordinated intra-subunit interactions and allosteric regulation[20,21]. Furthermore, the Rpd3S complex displayed a di-nucleosome preference with an optimal linker DNA length for H3K36me3-directed deacetylation[20–22]. Collectively, these biochemical features suggest multivalent and dynamic engagement of Rpd3S with its modified nucleosomal substrates, whose structural basis awaits in-depth investigation. Here, using cryo-electron microscopy (cryo-EM) and biochemical and yeast-based assays, we provide molecular and mechanistic insights into crosstalk between Rpd3S assembly, catalysis and modification at the open chromatin level.

## Overall architecture of the Rpd3S core complex

For structural analysis, we co-expressed full-length *S. cerevisiae* Rpd3, Sin3, Rco1, Eaf3 and Ume1 in a baculovirus–insect cell system and successfully reconstituted the Rpd3S holo-enzyme (Fig. 1a and Extended Data Fig. 1a). Next, we performed single-particle cryo-EM analysis and

[1]State Key Laboratory of Molecular Oncology, MOE Key Laboratory of Protein Sciences, SXMU-Tsinghua Collaborative Innovation Center for Frontier Medicine, School of Medicine, Tsinghua University, Beijing, China. [2]Beijing Frontier Research Center for Biological Structure and Beijing Advanced Innovation Center for Structural Biology, Beijing, China. [3]Shanghai Institute of Immunology, Shanghai Jiao Tong University School of Medicine, Shanghai, China. [4]Research Center of Basic Medicine, Academy of Medical Sciences, State Key Laboratory of Esophageal Cancer Prevention and Treatment, Zhengzhou University, Zhengzhou, China. [5]Technology Center for Protein Sciences, MOE Key Laboratory of Protein Sciences, School of Life Sciences, Tsinghua University, Beijing, China. [6]Department of Biochemistry and Molecular Cell Biology, Shanghai Key Laboratory for Tumor Microenvironment and Inflammation, MOE Key Laboratory of Cell Differentiation and Apoptosis, Shanghai Jiao Tong University School of Medicine, Shanghai, China. [7]State Key Laboratory of Membrane Biology, School of Life Sciences, Tsinghua University, Beijing, China. [8]Tsinghua-Peking Center for Life Sciences, Beijing, China. [9]These authors contributed equally: Haipeng Guan, Pei Wang. ✉e-mail: yancy2019@tsinghua.edu.cn; lht@tsinghua.edu.cn

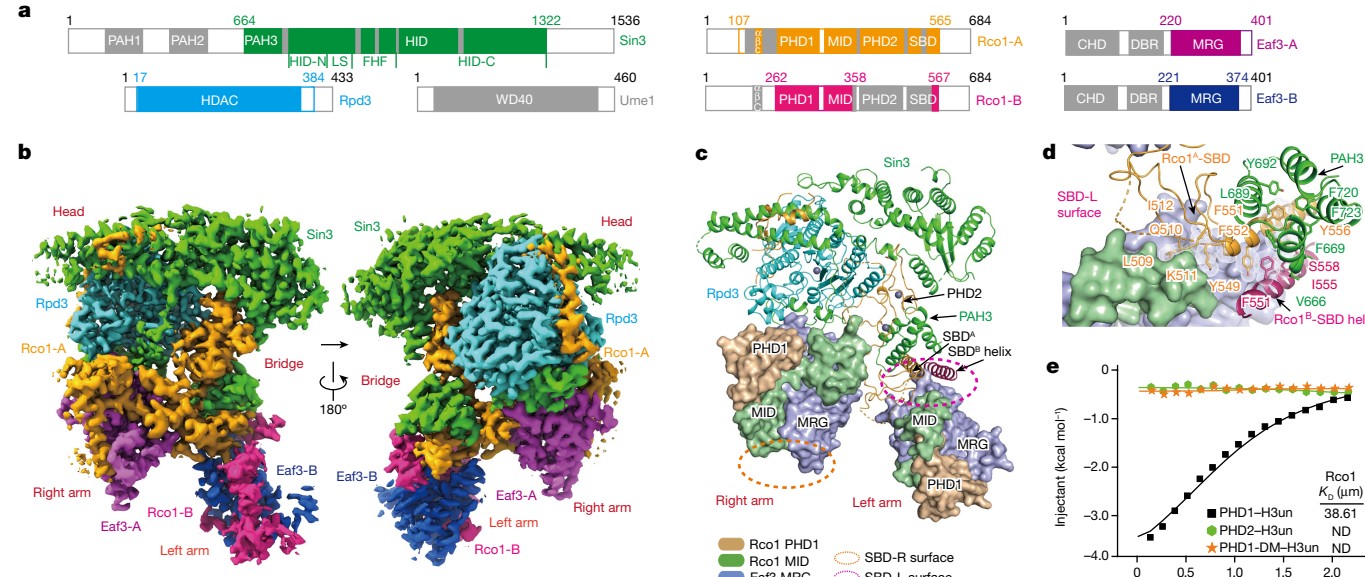

**Fig. 1 | Cryo-EM structure of Rpd3S complex and interactions among subunits. a**, Schematic representation of the domain organizations of the Rpd3S complex. The colour scheme for Rpd3S complex subunits is indicated in **b**. Domains that are not structurally resolved are coloured grey. **b**, Cryo-EM map of the entire Rpd3S complex integrated with a 2.7 Å map of the head–bridge–right arm region and a 3.2 Å map of the left arm region. WD40, WD40 β-propeller domain. **c**, Global view of the Rpd3S complex, highlighting the Rco1–Eaf3 heterodimers. The PHD1 (light brown) and MID (reseda) domains

of Rco1 and the MRG domain (blue violet) of Eaf3 are shown in surface representation; other domains are shown in cartoon form. The dotted lines represent the SBD surfaces (SBD-R and SBD-L) at different angles. **d**, Close-up view of interactions of the SBD domains of Rco1 and the PAH3 domain of Sin3 in the SBD-L surface and bridge region. **e**, Isothermal titration calorimetry fitting curves for indicated histone peptides with PHDs of Rco1. H3un, unmodified H3$_{1-10}$ peptide; PHD1-DM, PHD1 double mutant (E260K/D261K). ND, not determined.

acquired an overall structural map of Rpd3S at 3.2 Å resolution. Using a focused refinement strategy, we obtained 2.7 Å and 3.2 Å maps of two local regions, the head–bridge–right arm and the left arm of the Rpd3S complex, which enabled de novo model building of approximately 1,800 residues of Rpd3S components with high accuracy (Fig. 1b, Extended Data Fig. 2, Extended Data Table 1 and Supplementary Video 1).

In the structural model, we identified one copy of Rpd3 and Sin3, and two copies of Eaf3 and Rco1 (Fig. 1b and Extended Data Fig. 3a). Although Ume1 exists as a stoichiometric component of Rpd3S (Extended Data Fig. 1a), its map is missing. Cross-linking mass spectrometry (XL-MS) further confirmed the existence of Ume1, which mainly interacts with the C-terminal tail of Sin3 (Extended Data Fig. 1d). Both the Sin3 C-terminal tail and Ume1 are invisible in the map, suggesting their flexible disposition. The modelled structure contains residues 664–1322 of Sin3, 17–384 of Rpd3, 220–401 of Eaf3-A, 221–374 of Eaf3-B, 107–565 of Rco1-A and 262–358 of Rco1-B (Fig. 1a), which constitute a catalytic core complex with Rpd3 positioned at the centre. The overall architecture of the Rpd3S core complex can be divided into four parts: a head region consisting of the HDAC-interacting domain (HID) of Sin3 and full-length Rpd3; a bridge region consisting of the plant homeobox domain 2 (PHD2) and Sin3-binding domain (SBD) of Rco1-A and the paired amphipathic helix 3 (PAH3) domain of Sin3; and right and left arms each composed of the MORF-related gene (MRG) domain of Eaf3 (MRG$_{Eaf3}$) and the PHD1 domain and MRG-interacting domain (MID) of Rco1 (PHD1–MID$_{Rco1}$) (Fig. 1b, Extended Data Fig. 3a and Supplementary Video 1).

## Asymmetric assembly of two Eaf3–Rco1 dimers

It has been reported that the Rpd3S complex contains two copies of Rco1, which are critical for the integrity of the complex and nucleosomal substrate recognition[23]. However, our structural study also revealed the existence of two copies of Eaf3. Within Rpd3S, Eaf3 and Rco1 first form

a heterodimer primarily through MRG$_{Eaf3}$ and PHD1–MID$_{Rco1}$. Then, two heterodimers asymmetrically bind Rpd3 and Sin3 to form a six-subunit catalytic core complex (Fig. 1c). Eaf3 comprises three domains: CHD, the DNA-binding region (DBR) and MRG, and Rco1 comprises the PHD1, MID, PHD2 and SBD domains. The MID domain was previously defined as a Sin3-interacting domain (SID), which was supposed to interact with the PAH domain of Sin3[24]. However, here we reveal that this SID domain dominates MRG$_{Eaf3}$ engagement. Therefore, we renamed it MID. Meanwhile, our structural analysis identified an SBD that occurs after PHD2 of Rco1. This SBD is composed of a knotted coil followed by an α-helix and directly interacts with PAH3 of Sin3 (Fig. 1c and Extended Data Fig. 4a–d).

The two highly integrated MRG–PHD1–MID modules constitute the right and left arms of the Rpd3S complex, which use distinct surfaces for asymmetric Rpd3S assembly. On the left arm, MRG$_{Eaf3-B}$ and MID$_{Rco1-B}$ form the SBD-L surface that interacts with the Rco1-A SBD and subsequently with Sin3 PAH3 for assembly (Fig. 1c). By contrast, the MRG–PHD1–MID module in the right arm uses the opposite SBD-R surface that involves MRG$_{Eaf3-A}$ and the Rco1-A PHD1 domain for Rpd3S assembly (Fig. 1c). Surprisingly, the helix element of Rco1-B's C-terminal SBD domain also participates in SBD-L surface engagement despite the invisibility of the knotted coil motif. As shown in Fig. 1d, the SBD domain of Rco1-A (orange) and the SBD helix of Rco1-B (pink) weld PAH3$_{Sin3}$ and the left arm through extensive hydrophobic interactions. The occurrence of two adjacent SBD helices is supported by XL-MS studies (Extended Data Fig. 1d) and suggests an overall conformational distinction between the two Rco1–Eaf3 heterodimers. Rco1 has two PHD fingers. Our isothermal titration calorimetry titration assays confirmed unmodified histone H3 residues 1–10 (H3$_{1-10}$) readout by PHD1 with a measured $K_d$ of approximately 39 μM, but PHD2 displayed no H3$_{1-10}$ reader activity owing to histone surface blocking (Fig. 1e and Extended Data Fig. 4e–h). In fact, PHD2 is sandwiched by Rpd3 and Sin3 and functions more like an assembly module rather than a reader (Fig. 1c). In summary, the Rpd3S complex has two CHDs for H3K36me3 readout

and two PHD1s for unmodified H3K4 readout, which can cooperatively increase the nucleosome-binding multivalency.

## Rpd3 coordination in the Rpd3S core complex

Most Rpd3 were modelled, except for the flexible terminal tails, which enables us to examine the structural organization centred on Rpd3. As shown in Extended Data Fig. 3b, Rpd3 is wrapped around by discrete regions of Sin3, Rco1-A and Eaf3-A, including HID-N, loopS, four-helix-finger (FHF) and HID-C of Sin3 HID domain, PHD1 and PHD2 fingers of Rco1-A, and a C-terminal helix tail (αE) of Eaf3-A from the front surface, as well as an extended α−β-coil (αβC) motif of Rco1-A from the back surface. This highly coordinated intra-subunit assembly is dominated by extensive hydrogen bonds and electrostatic pairs, which, along with several featured hydrophobic contacts, notably involving HID-N and FHF, ensure a stable association of the Rpd3S catalytic core (Extended Data Fig. 5a–g).

In addition to Rpd3, the largest subunit, Sin3, also has a scaffolding role in Rpd3S. The N-terminal PAH1 and PAH2 domains of Sin3, which are not visible in the structure, have been proposed to interact with specific transcription factors[25,26]. Our structural studies characterized a third PAH3 domain of Sin3, which is responsible for Rco1 interaction and left arm assembly. Moreover, we have redefined an HID region (749–1322) that wraps around the catalytic centre of Rpd3 from the front (Fig. 1a and Extended Data Fig. 3b). A PAH4 domain (1143–1216) has been reported to exist within Sin3[27]. However, structural alignment has revealed that the proposed PAH4 does not adopt a PAH fold despite a four-helix composition (Extended Data Fig. 4i). In fact, it exists as an integrated part of the HID-C subdomain in the head region of the Rpd3S complex (Fig. 1b).

In mammals, class-I HDACs form distinct complexes such as SMRT–NCoR, NuRD and MiDAC, whose activities are regulated by inositol phosphate that binds to a basic pocket around the catalytic centre[28–30] (Extended Data Fig. 5h–k). As a class-I HDAC, yeast Rpd3 harbours a conserved basic surface. However, the activities of yeast Rpd3 complex and its mammalian SIN3A and SIN3B counterparts do not require inositol phosphate[28,30]. In the Rpd3S complex, we observed that a long α-helix of Sin3 (α1) within the FHF motif covers the basic surface of Rpd3 and inserts two Glu fingers (Glu811 and Glu812) into the inositol phosphate-binding pocket (Extended Data Fig. 3b). The two acidic residues are conserved among Sin3 orthologues (Extended Data Fig. 3e), which suggests a role of Sin3 in regulating Rpd3 catalytic activity and explains why the Sin3 family class-I HDAC complexes are exempt from inositol phosphate regulation.

## Structure of Rpd3S bound to H3K36me3 nucleosome

The Rpd3S complex functions mainly as a H3K36me3-dependent nucleosomal deacetylase[20,22]. Our observation of two copies of Eaf3 in the Rpd3S complex further underscores the importance of H3K36me3 readout in histone deacetylation at the nucleosome level. To explore the underlying molecular basis, we reconstituted the H3K36me3 modified 'designer' nucleosome and prepared cross-linked Rpd3S–nucleosome complex for cryo-EM studies. Two engagement modes ('close' and 'loose') of the Rpd3S–nucleosome complexes were acquired in global density maps of 4.0 and 4.0 Å, respectively. Focused refinement led to significantly improved local maps, with 2.8 Å for the CHD–H3K36me3 nucleosome, 3.3 Å for the Rpd3S complex of the close state, and 3.4 Å for the Rpd3S complex of the loose state (Extended Data Fig. 6).

The overall structure of the Rpd3S catalytic core is largely unchanged, except for poorer density of the left arm, probably owing to a lack of nucleosomal contact (Fig. 2a and Extended Data Fig. 7a). Of note, the two CHD domains of Eaf3 became visible in the presence of H3K36me3 nucleosome, with Eaf3-B CHD recognizing H3K36me3 at the DNA super-helical location (SHL) +1 site and Eaf3-A CHD bound to the H3K36me3 mark at the DNA SHL +7 site (Fig. 2b, Extended Data Fig. 7b and Supplementary Video 2). Although the two H3K36me3 marks are symmetrically related by a dyad axis, simultaneous readout of two methyl marks by Rpd3S leads to its asymmetric engagement with the H3K36me3 nucleosome. In this case, one Rpd3S associates with the nucleosome disc from one side by grabbing two H3K36me3 marks and positions its catalytic centre of Rpd3 next to the N-terminal tail of histone H4 for deacetylation (Fig. 2b). In both close and loose states, the two CHD domains are similarly anchored by the H3K36me3 marks (Fig. 2c). Structural alignment revealed rotational displacement of the Rpd3S catalytic core from the close to the loose states, in which the MID domain of Rco1-A within the right arm leaves the linker DNA next to SHL +7 in the loose state (Fig. 2c and Supplementary Videos 3 and 4). The existence of two engagement modes suggests dynamic association between the Rpd3S enzyme and its nucleosomal substrate, which may be beneficial for enzymatic turnover towards different acetylation sites occurred to histones H3 and H4 tails. Moreover, our structural study revealed that the catalytic centre of Rpd3S is far away from the H2A and H2B tails in the context of a nucleosome (Fig. 2d), consistent with a primary role of Rpd3S in H3 and H4 deacetylation.

## Engagement of H3K36me3 nucleosome by Rpd3S

Our structures revealed multivalent engagements between Rpd3S and H3K36me3 nucleosome, including two pairs of CHD$_{Eaf3}$–H3K36me3 recognition, MID$_{Rco1}$–linker DNA interaction, and Sin3 HID–nucleosomal DNA interaction, which collectively contribute to the positioning of the H4 N-terminal tail towards the catalytic centre of Rpd3 (Fig. 3a,b). The CHD$_{Eaf3}$ domain adopts a chromo barrel fold and binds to H3K36me3 as well as nucleosomal DNA (Fig. 3c). The trimethyllysine group of H3K36me3 is inserted into an aromatic cage formed by Y23, Y81, W84 and W88 of CHD$_{Eaf3}$, and is stabilized by cation−π and hydrophobic interactions. Moreover, a G83–W88 loop of CHD$_{Eaf3}$ contributed to DNA contacts through electrostatic (K85) and hydrogen bonding (G83) interactions (Fig. 3c and Extended Data Fig. 7c–e). The linker DNA is recognized by the S315–K321 segment within the Rco1-A MID, which is consistent with previous biochemical studies[21] (Fig. 3d). The nucleosomal DNA at SHL +2.5 is recognized by the HID-C region of Sin3 with multiple electrostatic or hydrogen bonding pairs involving residues Q937, K941, H1221, Q1222 and K1244 of Sin3 and backbone phosphates across the DNA minor groove (Fig. 3e). Upon engagement mode switch from the close to loose states, the H3K36me3 readout and SHL +2.5 DNA contacts act as anchoring sites to allow rotational displacement of Rpd3S, and the linker DNA contacts are temporally disrupted during the process, reflecting enzyme–substrate engagement dynamics (Supplementary Video 5).

## Readout-guided histone deacetylation by Rpd3S

To biochemically explore multivalent engagement-guided histone deacetylation by Rpd3S, we synthesized combinatorially modified histones H4(K5acK8acK12acK16ac) and H3(K9acK14acK18acK23acK27acK36me3) and reconstituted a designer nucleosome for enzymatic assays. We used acetylation site-specific antibodies to monitor the removal of a particular histone acetylation mark. The site specificity was validated by dot blot assays, and notably, all the antibodies could tolerate a hyperacetylated state of H3 or H4 (Extended Data Fig. 8a). We measured the deacetylation efficiency of the designer nucleosome substrate (500 nM) by titrating enzyme concentrations from 0 to 25.6 nM under optimized conditions. As shown in Fig. 4a (left), wild-type Rpd3S could effectively remove most H3 and H4 acetylation marks to various extents, except for H3K9ac. This is consistent with previous reports that Rpd3S could act as a HDAC for histones H3 and H4[20].

Our structural studies revealed that the PHD1 finger of Rco1-A is next to the catalytic centre of Rpd3S. Given that PHD1 is a reader of

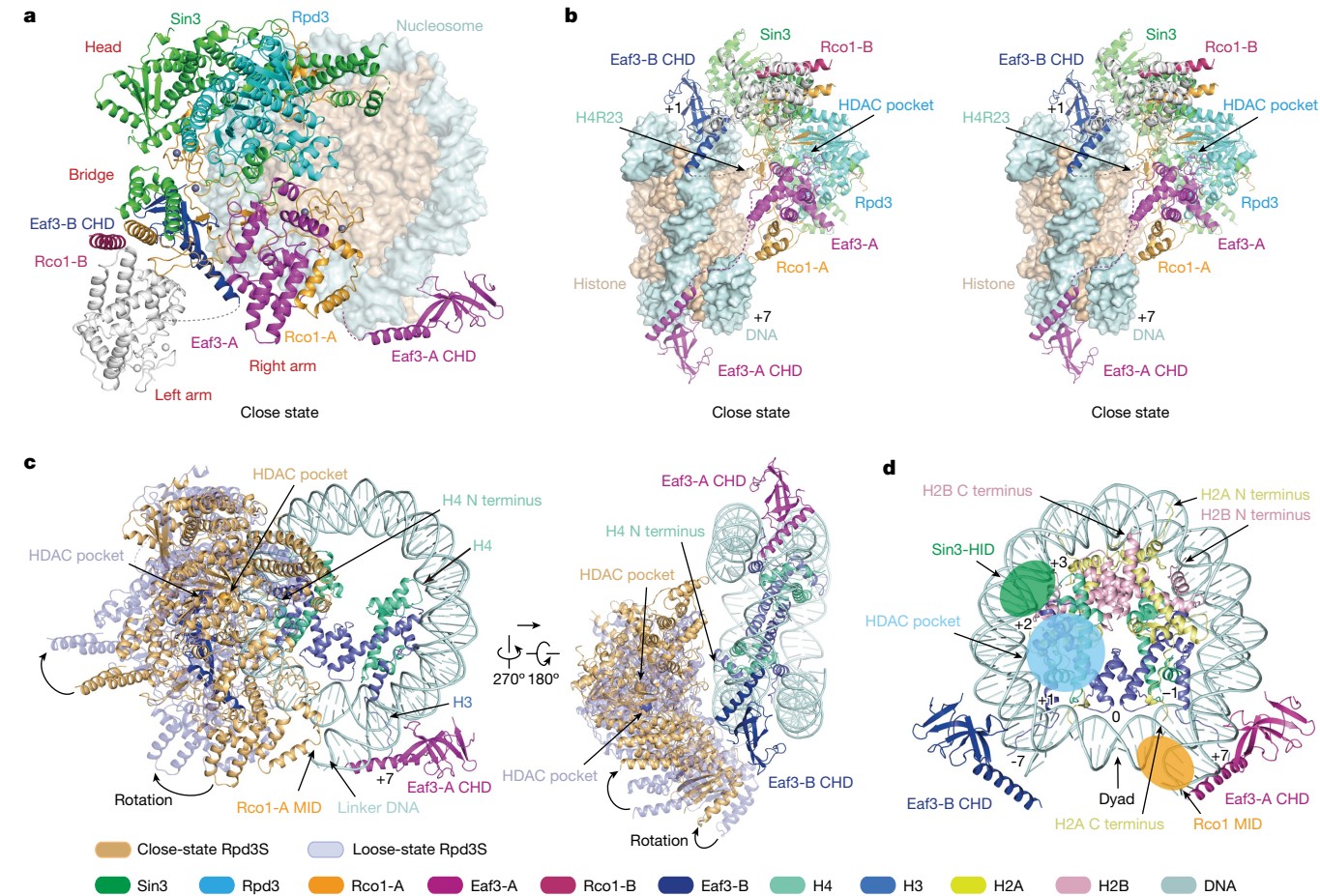

**Fig. 2 | Cryo-EM structure of Rpd3S complex bound to H3K36me3 nucleosome. a**, A model of the core Rpd3S complex bound to the H3K36me3 modified nucleosome in the close state. **b**, A stereo view of contact between the core Rpd3S complex and the H3K36me3 modified nucleosome in the close state. Sin3, Rpd3, Eaf3 and Rco1 are shown in cartoon form. The nucleosome is shown in surface representation. The invisible left arm region is coloured grey. **c**, Two views of the superimposition of close and loose state of core Rpd3S complex are shown in orange yellow and pale cyan, respectively. H3 and H4 of the histone octamer are highlighted in cartoon form. **d**, A model of the CHD domains with the H3K36me3 modified nucleosome. The tails of histone H2A and H2B are labelled with arrows. The positions of nucleosomal DNA are labelled as SHL positions.

unmodified histone H3K4 (Fig. 1e) and H3K14ac is one of the most preferred deacetylation sites (Fig. 4a, left), we docked a histone H3K14ac peptide onto Rpd3S based on homologous complex structures[30] of PHD–H3K4 and HDAC1–H4K16ac (Extended Data Fig. 4e). In our energy-minimized model (Fig. 4b,c), the histone H3K4 tail is anchored to the β1 surface of PHD1 such that H3K14ac is directed towards the catalytic centre of Rpd3 in an extended conformation. In this case, H3K9 is registered and consequently restricted at a place several residues away from the catalytic centre (Fig. 4d), which explains the observed low efficiency of H3K9ac deacetylation. Efficient deacetylation of H3K14ac underscores a crosstalk between unmodified H3K4 readout and H3 deacetylation. The promiscuous activities of Rpd3S towards histone H3 K18ac, K23ac, and K27ac suggest dynamic engagement of these sites with the catalytic pocket following combinatorial readout of H3K4 by Rco1-A PHD1 and of H3K36me3 by Eaf3-B CHD (Fig. 4b).

To explore the role of the PHD1 and CHD reader domains, we generated a Rco1(E260K/D261K) mutant that disrupted the reader activity of PHD1 (Fig. 1e and Extended Data Fig. 1b) and Rco1(L509A/Q510A/K511A/I512A/Y549A/Y556A/M560A) (Rco1(7mu)), which destabilized the SBD interface and consequently the left arm and Eaf3-B CHD (Fig. 1d and Extended Data Fig. 1c). Enzymatic assays revealed that, following PHD1 mutation, the deacetylation activities of Rpd3S towards histone H3, but not H4 were significantly affected (Fig. 4a, middle). Additionally, trimethylation of H3K4 had a similar effect on nucleosomal deacetylation as the E260K/D261K mutant (Extended Data Fig. 8b,d). This

is consistent with a previous report indicating that the presence of H3K4me3 negatively regulates Rpd3S activity and localization[19], and underscores the importance of unmodified H3K4 readout by PHD1 in H3 deacetylation. Meanwhile, the Rpd3S complex with Rco1(7mu) displayed severely compromised activities towards both H3 and H4 with a stronger effect on the latter (Fig. 4a, right). This result highlights an important role of the left arm, especially the Eaf3-B CHD domain, in coordinating nucleosomal substrate recognition and H3K36 methylation-guided deacetylation. Next, we investigated the effect of H3K36 methylation on Rpd3S activity by creating designer nucleosomes with unmodified H3K36 and poly-acetylated H3 and H4. Enzymatic assays revealed that H3K36me3 consistently enhances the activity of the Rpd3S complex across different sites, as observed in both wild-type and mutant samples (Extended Data Fig. 8d–f).

## In vivo modification crosstalk studies

To evaluate the importance of PHD1-guided H3 deacetylation by Rpd3S in yeast, we used a *STE11-HIS3* reporter system in which the HIS3 gene is fused to a naturally occurring cryptic promoter in the *STE11* gene[21]. As expected, the E260K/D261K reader mutant of Rco1 caused a marked cryptic transcription phenotype in spotting assays (Fig. 5a), suggesting in vivo functional defects of the Rpd3S mutant. Next, we examined changes in H3 and H4 acetylation levels between the wild-type strain and the PHD1 mutant. Consistent with our in vitro enzymatic assays,

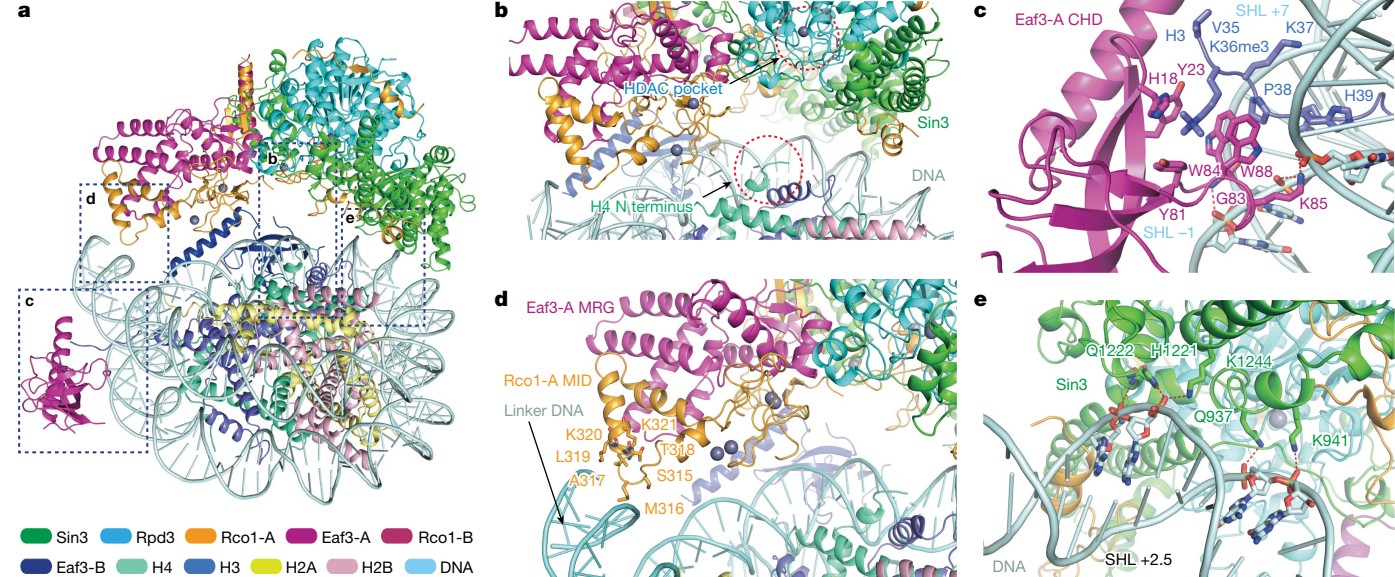

**Fig. 3 | Details of the interface between the Rpd3S complex and the H3K36me3 nucleosome. a**, A model of the core Rpd3S complex bound to the H3K36me3 modified nucleosome following histone H4 deacetylation. Interactions are outlined and shown in close up in the indicated panel. **b**, Detailed view of the location between the N terminus of H4 and Rpd3 in the close state as shown in **a**. **c**, Detailed view of the interactions between CHD and the H3K36me3 modified nucleosome as shown in **a**. The residues of CHD and the nucleotides of nucleosomal DNA involved in recognition and H3 tail

residues are shown as sticks. Selected hydrogen bonds are shown as red dashed lines. **d**, Detailed view of interactions between the Rco1 MID and linker DNA of nucleosome as shown in **a**. Residues at the interface are depicted as sticks. **e**, Detailed view of interactions between Sin3 and nucleosomal DNA at SHL +2.5 as shown in **a**. The residues of Sin3 and the nucleotides of nucleosomal DNA involved in recognition are shown as sticks. Selected hydrogen bonds are shown as red dashed lines.

the acetylation levels of histone H3 but not histone H4 were affected by the PHD1 mutant (Fig. 5b). These results confirmed a role for PHD1 in H3-specific deacetylation and highlighted the contribution of H3 acetylation in permitting cryptic transcription. Meanwhile, Rpd3S(7mu) exhibited a similar cryptic transcription phenotype and yielded consistent results in vivo compared to the in vitro enzymatic assays (Fig. 5a,b).

H3K9 remained largely acetylated following Rpd3S treatment in our designer nucleosome-based deacetylation assays (Fig. 4a). We reasoned that H3K9ac may serve as an anchoring mark for the reestablishment of H3 and H4 acetylation, given that both NuA3 and NuA4 histone acetyltransferase complexes contain an H3K9ac reader subunit—Taf14 and Yaf9, respectively[31,32]. We generated a H3K9R mutant strain

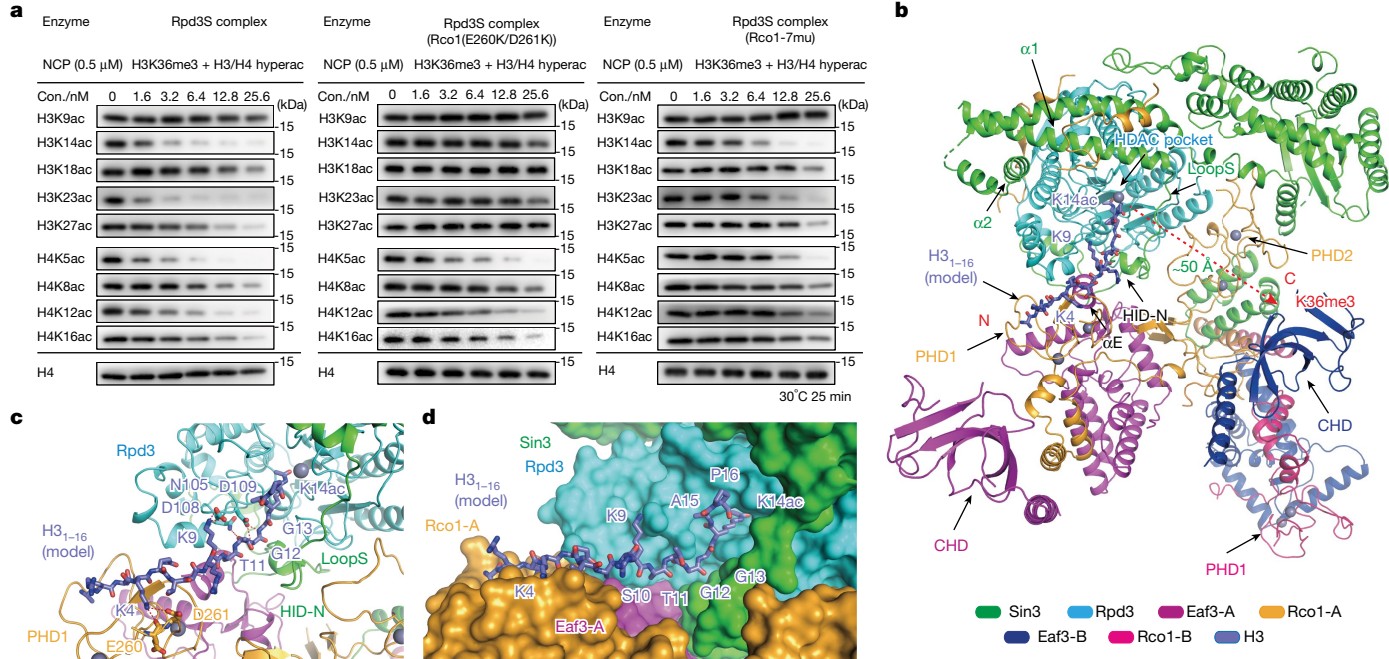

**Fig. 4 | Combinatorial readout-guided histone deacetylation by Rpd3S complexes. a**, A representative HDAC assay measuring activity of Rpd3S complexes containing wild-type Rco1 (left), Rco1 PHD1 mutants (middle) and Rco1 left arm region mutants (right) on H3K36me3 and hyperacetylated (hyperac) nucleosome. The reaction products were identified by western blot.

Data are representative of three independent experiments. **b**, Global view of the docking model of the Rpd3S complex with histone H3[1-16]K14ac on histone H3 deacetylation. **c,d**, Detailed views of interactions between the Rpd3S complex and histone H3K14ac in cartoon (**c**) and surface (**d**) representation.

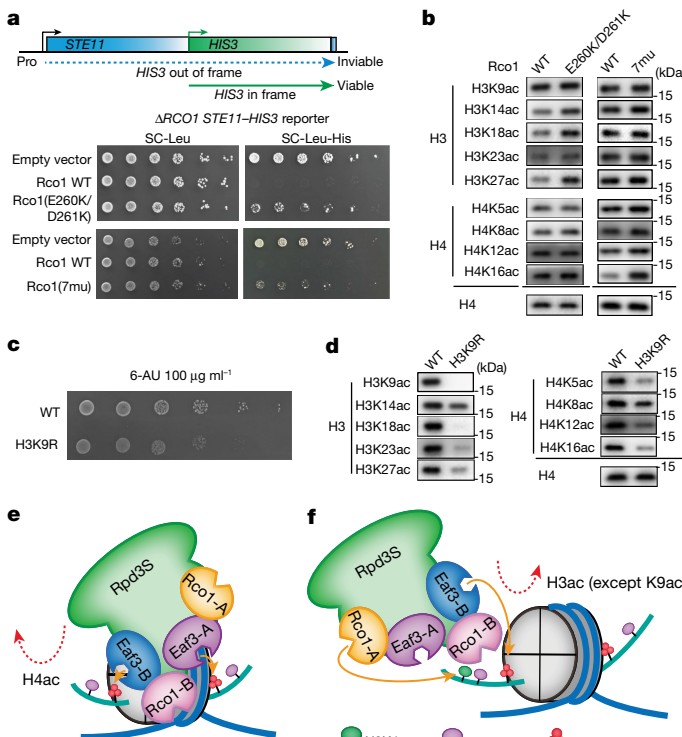

**Fig. 5 | In vivo modification crosstalk studies. a**, The test cryptic transcription phenotype caused by Rco1(E260K/D261K) or Rco1(7mu) mutants in a *STE11-HIS3* reporter strain (YBL853). WT, wild type. **b**, Western blot showing H3 and H4 acetylation levels at different sites in Rco1 wild type, E260K/D261K and 7mu mutants. **c**, Growth of H3 wild type and H3K9R mutant yeast. Fivefold serial dilutions of H3 wild type and H3K9R mutant cells were spotted onto plates with synthetic complete medium (with glucose) containing 100 μg ml⁻¹ 6-azauracil (6-AU) and cultivated at 30 °C for 3 days. **d**, Western blot showing global H3 and H4 acetylation defects in the H3K9R mutant yeast strain. **a**–**d**, One representative example of three independent experiments. **e**,**f**, Models of the Rpd3S complex bound to the H3K36me3 modified nucleosome on histone H4 deacetylation (**e**) and on histone H3 deacetylation (**f**).

to investigate the role of H3K9ac. The H3K9R mutant caused clear growth defects in spotting assays (Fig. 5c). We further observed global acetylation defects of histone H3 and H4 (Fig. 5d), suggesting that H3K9ac-mediated modification crosstalk may indeed have a role in promoting hyperacetylation of H3 and H4. Notably, our enzymatic assays revealed that the decreased acetylation of H3 and H4 is not the result of enhanced enzymatic activity of Rpd3S. Instead, we observed slightly impaired deacetylation activity of Rpd3S due to the K9R mutation (Extended Data Fig. 8c,d).

On the basis of our cryo-EM, biochemical, and functional studies, we propose the following modification crosstalk model for the Rpd3S regulatory axis. Histone H3K36me3 dual-mark readout by Rpd3S over a single nucleosome positions the catalytic centre of Rpd3 next to the histone H4 N-terminal tail for efficient deacetylation (Fig. 5e). By contrast, coordinated readout of H3K4un by Rco1-A PHD1 and of H3K36me3 by Eaf3-B CHD over one histone H3 tail determined the deacetylation activity of Rpd3S toward histone H3 (Fig. 5f). Moreover, the Rpd3S-resistant H3K9ac mark may raise another level of modification crosstalk to restore chromatin hyperacetylation.

## Discussion

Since the discovery of Rpd3 as a HDAC in 1996 (refs. 15,33), subsequent biochemical studies have revealed delicate regulatory mechanisms of the multi-subunit Rpd3 complexes[18,22]. Here we present the cryo-EM

structures of Rpd3S in its free and designer nucleosome-bound states, which together with our enzymatic and yeast genetics studies, enable a molecular dissection of Rpd3S complex assembly, its substrate engagement and enzymatic regulation.

The identification of two CHD-containing Eaf3 subunits was unexpected, expanding a layer of previously unappreciated multivalency for Rpd3S–nucleosome engagement. Structural studies revealed that two CHD_Eaf3 domains cooperatively bind to a fully methylated H3K36me3 nucleosome, which oriented Rpd3S over a nucleosomal disc for histone H4-specific deacetylation. Efficient histone deacetylation of the H4 tail depends on full methylation of H3K36me3 over a single nucleosome. Disruption of H3K36me3 readout by one CHD_Eaf3 reader largely impaired the deacetylation efficiency of histone H4 but not the H3 tail (Fig. 4a, right). Full methylation of nucleosomal H3K36me3 probably indicates a chromatin state of prolonged active elongation after multiple cycles of Set2-mediated H3K36 methylation. Thus, the occurrence of a full H3K36me3 state, but not its hemimethylated form, may serve as an attenuation 'checkpoint' for Rpd3S-mediated H4 deacetylation.

Besides the H3K36me3–H3K36me3–H4ac mode of 'reading–erasing' crosstalk, our structural and mutagenesis analyses further suggested a H3K4un–H3K36me3–H3ac mode of crosstalk accomplished by combinatorial actions of PHD1_Rco1 and CHD_Eaf3. Notably, our complex structural studies revealed that the catalytic centre of Rpd3 is positioned away from the H2A and H2B tails upon methylated nucleosome engagement. By contrast, a recent structural study of the yeast HDA1 complex, a class-II HDAC, revealed that its catalytic centre is positioned next to the H2B tail in the nucleosomal context, therefore acting as an H2Bac eraser[34]. Collectively, these findings highlight the importance of different modes of multivalent engagement in achieving catalytic specificity towards different histone tails by a multi-subunit deacetylase complex.

Rpd3S deacetylates multiple acetylation sites on H3 and H4, except for H3K9ac (Fig. 4a, left and Extended Data Fig. 9a). It efficiently targets H3K14ac and H3K23ac, which are primary products of the NuA3 acetyltransferase that regulates transcriptional elongation by antagonizing Rpd3S[35]. NuA3 contains reader modules, such as the TAF14 YEATS domain for H3K9ac readout and the Pdp3 PWWP domain for H3K36me3 readout[36]. Of note, NuA4, the histone H4 acetyltransferase in yeast, also contains an H3K9ac reader subunit—Yaf9[32]—and shares the H3K36me3 reader subunit Eaf3 with Rpd3S[31,37]. Conceivably, H3K9ac and H3K36me3 may serve as 'seed' marks for NuA3–NuA4 recruitment and the reestablishment of a hyperacetylated H3 or H4. Therefore, Rpd3S and NuA3–NuA4 act as counteracting 'eraser–writer' enzymatic pairs to optimize gene expression dynamics during transcriptional elongation and beyond (Extended Data Fig. 9b).

Deleting Set2 in yeast leads to an increase in histone exchange and the accumulation of H3K56ac over transcribed genes[38]. However, H3K56ac within a nucleosome cannot be erased by Rpd3S owing to its buried nature and the need for an intact nucleosome for Rpd3S engagement. This finding is consistent with previous functional studies[39] and our yeast knockout experiments (Extended Data Fig. 9c). Rtt109, in partnership with histone chaperones Asf1 and Vps75, serves as a writer for H3K56ac and H3K9ac, regulating histone exchange[40]. The inability of Rpd3S to deacetylate H3K56ac and H3K9ac suggests that Rpd3S does not directly downregulate histone exchange by counteracting Rtt109. Set2 methylation of H3K36 is likely to indirectly suppress histone exchange by inhibiting cryptic transcription following H3K36me3-mediated H3 and H4 deacetylation by Rpd3S.

Previous studies suggest that Rpd3S activity can be enhanced by a di-nucleosome context[20]. Our structural studies show that the Rpd3S complex primarily binds to a single nucleosome through readout of dual H3K36me3 mark. Modelling studies suggest that although the CHD domains of Eaf3-A and Eaf3-B—along with Rco1-A PHD1—recognize one nucleosome, the PHD1 finger of Rco1-B is well-positioned for unmodified histone H3K4 readout of an adjacent nucleosome approximately 40 bp away (Extended Data Fig. 9d). This may suggest a mechanism by

which Rpd3S spreads from one nucleosome to another for efficient deacetylation. Alternatively, when the stoichiometry of Rpd3S to nucleosome is high, two Rpd3S molecules may simultaneously bind to two adjacent nucleosomes (Extended Data Fig. 9e,f), cooperatively recognizing a di-nucleosome unit with optimal linker DNA length[22], such as 30–40 bp, for catalysis. Further structural studies are needed to fully understand the recognition mechanism of di-nucleosome engagement by Rpd3S.

Epigenetic regulation ensures complexities from genes to phenotypic traits, and the evolution of multi-subunit epigenetic machineries represents a molecular strategy to afford the regulatory potential. Beyond Rpd3S and Rpd3L in yeast, a diverse family of HDAC complexes have been characterized in high eukaryotes to exert versatile physiological function, and their dysregulation is often linked to human disease. By offering a molecular visualization of a prototype class-I HDAC complex in chromatin context-dependent gene regulation, the studies described here lay a framework for better understanding versatile class-I HDAC complexes in eukaryotic species and pave the way for structure- and mechanism-based development of HDAC activity-modifying drugs.

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

## Methods

### Protein expression and purification

The components of the Rpd3S complex, including full-length Rpd3, Sin3, Rco1, Ume1, and Eaf3, were amplified from total yeast DNA by PCR. All of these components and their mutants were cloned into modified pFastbac baculoviral expression vectors. A TEV-cleavable Strep-tag was engineered at the N terminus of Rco1 for affinity purification. The bac-to-bac baculovirus system was used to express the target protein in insect cells. Baculovirus of all components of Rpd3S were mixed in a predetermined ratio to achieve stoichiometric expression in cells. Sf9 insect cells were co-transfected at 27 °C for 72 h. Cells were harvested by centrifugation at 2,700 rpm for 20 min at 4 °C, and the pellet was resuspended in cell lysis buffer (20 mM Tris-HCl pH 7.5, 300 mM NaCl, 5% glycerol, 0.1% NP-40) supplied with 1 mM PMSF and a protease inhibitor cocktail. Cells in the lysis buffer were sonicated using 56% amplification with 3 s on and 7 s pulse for 15 min on ice. Cell debris were removed by centrifugation at 4 °C, 25,000 rpm for 1 h. The cleared lysate was co-incubated with strep beads and then successively washed by cell lysis buffer, and the target protein was finally eluted by elution buffer (100 mM Tris-HCl pH 8.0, 150 mM NaCl, 10 mM D-desthiobiotin). Fractions containing Rpd3S were incubated overnight at 4 °C with TEV protease to cleave off the tag. The digested protein was further purified with an anion-exchange column (GE Healthcare). Target proteins were collected, concentrated and purified by size-exclusion chromatography using a Superose 6 Increase 5/150 GL column (GE Healthcare) in buffer (20 mM HEPES pH 7.5, 150 mM NaCl).

### Nucleosome reconstitution

Widom 601 sequence and a 20-bp linker DNA emitting from one side were purified from a plasmid encoding 20 repeats (each flanked by the EcoRV restriction enzyme cutting site) of the sequence as previously described[41]. The full-length unmodified *Xenopus laevis* histones H2A, H2B, H3 and H4 were expressed in *Escherichia coli* strain BL21 (DE3) and purified using a previously reported method[42]. All modified H3 or H4 were synthesized by KS-V Peptide (KS-V Peptide). Histone octamers containing either unmodified, K36me3 H3, or acetylated H3/H4 were refolded as previously described[41]. A modification of the salt gradient method described by Thomas and Butlerz[43] was used for the reconstitution of histone octamer with DNA. Histone octamer and DNA were combined at 2 M NaCl; the gradual reduction of the salt concentration to 0.25 M NaCl over a period of 36 h led to the formation of nucleosome core particles (NCPs).

### Sample preparation and cryo-EM data collection

The Rpd3S–NCP complex was obtained by mixing 15 μM protein with 5 μM NCP. We purified and stabilized the Rpd3S and Rpd3S–NCP complexes using the GraFix method[44]. To form the gradient, 6 ml top solution containing 50 mM NaCl, 20 mM HEPES (pH 7.5), and 10% glycerol (Sigma) was added to a tube (Beckman, 331372). The bottom solution (6 ml) containing 50 mM NaCl, 20 mM HEPES (pH 7.5), 30% glycerol, and 0.15% glutaraldehyde was then injected into the bottom of the tube using a syringe with a blunt-ended needle. The tubes were placed into a gradient master (BioComp) to form a continuous density and glutaraldehyde gradient. Finally, 200 μl of sample was loaded. The sample tubes were ultracentrifuged at 4 °C for 14 h at a speed of 35,000 rpm. (Beckman, Rotor SW-41Ti). Fractions were collected every 500 μl. The best fractions were selected and dialysed to buffer (20 mM HEPES pH 7.5, 50 mM NaCl, 3% glycerol) for electron microscopy sample preparation. For cryo-EM, an aliquot of 4 μl of the sample at a concentration of ~0.2 mg ml⁻¹ was applied to glow-discharged grids (Quantifoil 1.2/1.3). The grids were then blotted for 3.5 s and plunged into liquid ethane cooled by liquid nitrogen, using a Vitrobot (FEI). The EM grids were imaged on a Titan Krios transmission electron microscope (FEI) operated at 300 kV. A total of 6,990 images of the Rpd3S complex were collected on K3 (Gatan) with

a pixel size of 0.55 Å per pixel, and a total of 10,786 images of Rpd3S–nucleosome were collected on K3 (Gatan) with a pixel size of 0.541 Å per pixel. AutoEMation2 was used for automated data collection[45]. Defocus was set automatically, with values ranging from −1.5 to −1.8 μm. The fluence was ~50 e⁻ Å⁻² fractionated into 32 frames (exposure time 2.56 s).

### Image processing

Motion correction was performed using the MotionCor2[46]. CTF parameters were estimated using GCTF[47]. For the Rpd3S complex dataset, after automatic particle picking and 3 rounds of 2D classification in RELION[48,49], ~2.63 million particles were selected for the first round of 3D classification. The initial model was generated by RELION and low-pass-filtered to 60 Å for a 3D reference. After 2 rounds of 3D classification, ~670,000 particles were selected and used for 3D auto-refinement and post-processing, resulting in a reconstruction of the entire Rpd3S complex at 3.2 Å. Following local mask 3D classification for head–bridge–right arm, ~348,000 particles were selected and 3D auto-refined and post-processed, resulting in a local density map of the head–bridge–right arm with a resolution of 2.7 Å. Similarly, the local density map of the bridge–left arm was reconstructed at 3.2 Å. For the Rpd3S–nucleosome complex datasets, two datasets of 10,786 micrographs were selected. After automatic particle picking and 2 rounds of 2D classification in RELION, ~2.59 million particles were selected for 3D classification. The initial model was generated by RELION and low-pass-filtered to 60 Å for a 3D reference. After 2 rounds of 3D classification, two representative classes of the Rpd3S–nucleosome complex were relatively better distinguished with the change of the Rpd3S complex on the nucleosome. For the CHD–nucleosome complex, ~427,000 particles were used to 3D auto-refine and post-process in a close state, resulting in a local density map of the CHD–nucleosome complex with a resolution of 2.8 Å. After one round of 3D classification on the Rpd3S complex, two classes were performed, resulting in two distinct states of stable Rpd3S complexes. Both classes were subject to global 3D refinement, yielding two density maps at resolutions of 4.0 Å (close state), 4.0 Å (loose state). After 3D refinement of Rpd3S complex, a 3.3 Å map and a 3.4 Å map of the Rpd3S in two classes were resolved.

The local resolution map was created using RELION[49] and represented using UCSF Chimera[50]. All reported resolutions are based on the gold-standard Fourier shell correlation (FSC) 0.143. The final FSC curves were corrected for the effect of a soft mask with high-resolution noise substitution.

### Model building

The whole structural model of the Rpd3S complex was manually built in COOT[51] according to the 2.7 Å map of the head–bridge–right arm region and a 3.2 Å map of the bridge–left arm region. AlphaFold was used to predict and determine the special knotted coil in Rco1[52]. The structures of the nucleosome (PDB ID: 6ESF) and Eaf3 CHD (PDB ID: 3E9G) served as initial structural templates for the CHD–nucleosome model, which was docked into the cryo-EM maps using UCSF Chimera[50]. The Eaf3-A CHD was built and adjusted in COOT, while Eaf3-B CHD was docked into the cryo-EM map. Two structural models of the Rpd3S–nucleosome complex were built by docking the Rpd3S complex and CHD–nucleosome structures into a 2.8 Å map of CHD–nucleosome, a 3.3 Å map and a 3.4 Å map of the Rpd3S complex, respectively in two states, followed by rigid-body fitting and manual model building. The models were refined in real space using Phenix[53]. Statistics of the map reconstruction and model refinement are shown in Extended Data Tables 1 and 2. The final models were evaluated using MolProbity[54]. Map and model representations in the figures were prepared by PyMOL (https://pymol.org/), UCSF Chimera[50] or UCSF ChimeraX[55].

### Isothermal titration calorimetry

All calorimetric experiments on the wild-type or mutant PHD1 domain of Rco1 proteins were conducted at 25 °C using a MicroCal PEAQ-ITC

instrument (Malvern Panalytical). All proteins and synthetic histone peptides were prepared under the same titration buffer containing 20 mM Tris 7.5, 50 mM NaCl, and 5% glycerol. The protein concentration was determined by absorbance spectroscopy at 280 nm. Peptides (>95% purity) were quantified by weighing on a large scale, aliquoted, and freeze-dried for individual use. Acquired calorimetric titration curves were analysed using Origin 7.0 (OriginLab) with the 'one set of binding sites' fitting model. The detailed peptide sequence information is $H3_{1-10}$un: ARTKQTARKS.

### Antibodies
Antibodies used: rabbit polyclonal anti-H3K9ac (ABclonal, A7255, dilution: 1:5,000), mouse monoclonal anti-H3K14ac (PTM BIO, PTM-157, clone name: 1A4, dilution: 1:500), mouse monoclonal anti-H3K18ac (PTM BIO, PTM-158, clone name: 9E1, dilution: 1:750), rabbit polyclonal anti-H3K23ac (PTM BIO, PTM-115, dilution: 1:1,500), mouse monoclonal anti-H3K27ac (PTM BIO, PTM-160, clone name: 12G5, dilution: 1:2,000), mouse monoclonal anti-H3K56ac (PTM BIO, PTM-162, clone name: 9G9, dilution: 1:1,000), rabbit polyclonal anti-H3K36me3 (PTM BIO, PTM-625, dilution: 1:1,000), rabbit monoclonal anti-H4K5ac (Sigma-Aldrich, 04-118, clone name: RM156, dilution:1:5,000), rabbit polyclonal anti-H4K8ac (ABclonal, A7258, dilution: 1:6,000), rabbit polyclonal anti-H4K12ac (ABclonal, A22754, clone name: ARC56881, dilution: 1:1,000), rabbit monoclonal anti-H4K16ac (Cell Signaling Technology, 13534S, clone name: E2B8W, dilution:1:500), mouse monoclonal anti-H4 (PTM BIO, PTM-1009, clone name: 3F2, dilution: 1:1,000), and rabbit polyclonal anti-H3 (ABclonal, A2348, dilution: 1:2,000).

### HDAC assays
For the Rpd3S deacetylation assays, H3Kac, H4Kac and H3K36me3 nucleosomes (500 nM) assembled in vitro were treated with different Rpd3S complexes at different concentrations in a buffer with 20 mM HEPES 7.5, 150 mM NaCl, and 0.2 mg ml$^{-1}$ BSA at 30 °C. For the differences in enzyme activity caused by different histone modifications and H3K9R mutant, we adopted the enzyme activity assays in a buffer with 20 mM HEPES 7.5, 150 mM NaCl, 0.2 mg ml$^{-1}$ BSA and 0.15 µg µl$^{-1}$ salmon sperm DNA at 30 °C. For nucleosome samples, 80 mM EDTA and 5× SDS–PAGE gel loading buffer were added after 25 min. The samples were boiled for 5 min at 95 °C and resolved by 4–20% SDS–PAGE. After transferring to PVDF membrane, H3K9ac, H3K14ac, H3K18ac, H3K23ac, H3K27ac, H4K5ac, H4K8ac, H4K12ac, H4K16ac and total H4 were detected by western blot with specific antibodies on separate gels. Western blot bands were visualized by ECL.

### Cross-linking mass spectrometry
The purified Rpd3S complex was cross-linked with 5 mM bis (sulfosuccinimidyl) suberate (BS3) at room temperature for 2 h. The reaction was quenched with 40 mM Tris-HCl pH 7.5. The cross-linked sample was then excised for in-gel digestion and identified by mass spectrometry. To begin the in-gel digestion process, the sample was disulfide-reduced with 25 mM dithiothreitol (DTT) and alkylated with 55 mM iodoacetamide. In-gel digestion was performed using sequencing grade-modified trypsin in 50 mM ammonium bicarbonate at 37 °C overnight. The peptides were extracted twice with 1% trifluoroacetic acid in a 50% acetonitrile aqueous solution for 30 min. The peptide extracts were then centrifuged in a SpeedVac to reduce the volume.

For liquid chromatography–tandem mass spectrometry (LC–MS/MS) analysis, peptides were separated by a 60 min gradient elution at a flow rate 0.300 µl min$^{-1}$ with a Thermo-Dionex Ultimate 3000 HPLC system, which was directly interfaced with the Thermo Orbitrap Fusion mass spectrometer. The analytical column was a homemade fused silica capillary column (75 µm internal diameter, 150 mm length; Upchurch) packed with C-18 resin (300 A, 5 µm; Varian). Mobile phase A consisted of 0.1% formic acid, and mobile phase B consisted of 100%

acetonitrile and 0.1% formic acid. The Orbitrap Fusion mass spectrometer was operated in the the data-dependent acquisition mode using Xcalibur3.0 software, and there was a single full-scan mass spectrum in the Orbitrap (350–1,550 $m/z$, 120,000 resolution) followed by 3 s data-dependent MS/MS scans in an Ion Routing Multipole at 30% normalized collision energy. The MS/MS spectra from each LC–MS/MS run were searched against the selected database using the Proteome Discovery searching algorithm (version 1.4). Raw data were processed with pLink2 software[56]. A crosslink network diagram was prepared using xiNET[57].

### Computational docking of $H3_{1-16}K14ac$ into the Rpd3S complex
The readouts of histone H3K4un by the PHD1 of Rco1 and PHD of BHC80 are highly conserved. Additionally, Rpd3 is highly conserved with HDAC1. Based on this information, structures of the Rpd3S and Rpd3S–H3K36me3 nucleosome complexes, BHC80–PHD (PDB ID: 2PUY) and HDAC1–H4K16Hx (PDB ID: 5ICN) were prepared for docking using the protein preparation wizard in Maestro (Schrödinger, release 2018-1). Hydrogens were added and the protonation states of titratable amino acids were determined during the protein preparation. Docking was then performed using GLIDE/SP-peptide in Schrödinger[58]. Histone $H3_{1-7}K4un$ was docked into the PHD1 of Rco1-A and histone $H3_{12-16}K14ac$ was docked into the HDAC catalytic pocket of Rpd3. Simultaneously, $H3_{8-11}$ was stretched out between $H3_{1-7}K4un$ and $H3_{12-16}K14ac$, hindered by hydrogen bond interactions and steric hindrance.

### Spotting assays
The Rco1 and mutants, along with their native promoters, were cloned onto pRS415 plasmids. The plasmids (including control plasmid) were then transformed into the *STE11-HIS3* reporter strain (YBL853). To analyse the growth of the yeast strains, fivefold serial dilutions of fresh culture concentrated to an optical density at 600 nm ($OD_{600}$) of 0.4 were spotted onto the SC-Leu (control) and SC-His-Leu plates until saturation.

To analyse the importance of H3K9ac on yeast growth, we used a CRISPR–Cas9 genome editing method[59] to target the *HHT1* and *HHT2* genes and obtain the H3K9R mutant in the W303-1a strain. The cells were diluted to an $OD_{600}$ of 0.4 and fivefold serially diluted. About 5 µl of each dilution was spotted on a 100 µg ml$^{-1}$ 6-AU plate until saturation.

### Western blotting
To examine the levels of acetylation caused by Rco1 mutants in vivo, the Rco1 wild type and mutants, along with their native promoters, were cloned to pRS415 plasmids. The plasmids were then transformed into an Rco1-deleted strain (YBL534). The yeast strains were grown overnight at 30 °C in Leu medium, diluted to an $OD_{600}$ of 0.1, and grown for another 8 h to an $OD_{600}$ of 0.8–1.0. To examine the changes of H3 and H4 acetylation in an H3K9R mutant strain, the W303-1a and W303-1a-H3K9R were grown overnight at 30 °C in YPD, diluted to an $OD_{600}$ of 0.1, and grown for another 8 h to an $OD_{600}$ of 0.8–1.0. Yeast cell protein was extracted as described[60]. The samples were boiled for 20 min at 95 °C and resolved by 4–20% SDS–PAGE. After transferring to a PVDF membrane, H3K9ac, H3K14ac, H3K18ac, H3K23ac, H3K27ac, H3K56ac, H4K5ac, H4K8ac, H4K12ac, H4K16ac and total H4 were detected by western blot with specific antibodies on separate gels. Western blot bands were visualized by ECL.

To examine the levels of H3K56ac caused by Rpd3S complex in vivo, the pRS415-Rco1 and control plasmid were then transformed into the YBL534 strain. The yeast strains were grown overnight at 30 °C in Leu medium, diluted to an $OD_{600}$ of 0.1, and grown for another 8 h to an $OD_{600}$ of 0.8–1.0. H3K9ac, H3K14ac, H3K18ac, H3K23ac, H3K27ac, H3K56ac, and total H3 were detected by western blot with specific antibodies on separate gels. Western blot bands were visualized by ECL.

All yeast strains were constructed using standard procedures and are listed in Supplementary Fig. 1.

## Reporting summary

Further information on research design is available in the Nature Portfolio Reporting Summary linked to this article.

## Data availability

Cryo-EM maps have been deposited in the Electron Microscopy Data Bank (EMDB) under accession numbers EMD-33845 (whole Rpd3S complex), EMD-33846 (head–bridge–right arm), EMD-33847 (bridge–left arm), EMD-33848 (Eaf3 CHD bound to H3K36me3 nucleosome), EMD-33849 (Rpd3S complex in loose state), EMD-33850 (Rpd3S complex in close state), EMD-33851 (Rpd3S complex bound to H3K36me3 nucleosome in close state) and EMD-33852 (Rpd3S complex bound to H3K36me3 nucleosome in loose state). Atomic coordinates have been deposited in the Protein Data Bank under accession numbers 7YI0 (whole Rpd3S complex), 7YI1 (Eaf3 CHD bound to H3K36me3 nucleosome), 7YI2 (Rpd3S complex in loose state), 7YI3 (Rpd3S complex in close state), 7YI4 (Rpd3S complex bound to H3K36me3 nucleosome in close state) and 7YI5 (Rpd3S complex bound to H3K36me3 nucleosome in loose state).

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

**Acknowledgements** We thank X. Li (Tsinghua University) and F. Yang (Tsinghua University) for technical support during EM image acquisition. We thank the Tsinghua University Branch of China National Center for Protein Sciences (Beijing) for providing the cryo-EM facility support. We appreciate the peptide synthesized and provided by KS-V Peptide Biological Technology (Hefei, China). We thank our funding sources: National Key R&D Program of China (2021YFA1300100 to H.L. and B.L.; 2020YFA0803303 to H.L.); National Natural Science Foundation of China (92153302 and 32230019 to H.L.; 32030019 to B.L.); and the Fundamental Research Funds for the Central Universities and Shanghai Frontiers Science Center of Cellular Homeostasis and Human Diseases (to B.L.).

**Author contributions** H.L. conceived the project. H.G. and P.W. performed cryo-EM sample preparation, data collection, and biochemical and functional studies with help from P.Z., C.R., Y.O., B.P., X.Z. and J.L. H.G., H.L. and C.Y. performed electron microscopy analyses and model building. H.L., H.G. and P.W. analysed the structures and wrote the manuscript with input from B.L. and C.Y.

**Competing interests** The authors declare no competing interests.

**Additional information**
**Correspondence and requests for materials** should be addressed to Chuangye Yan or Haitao Li.

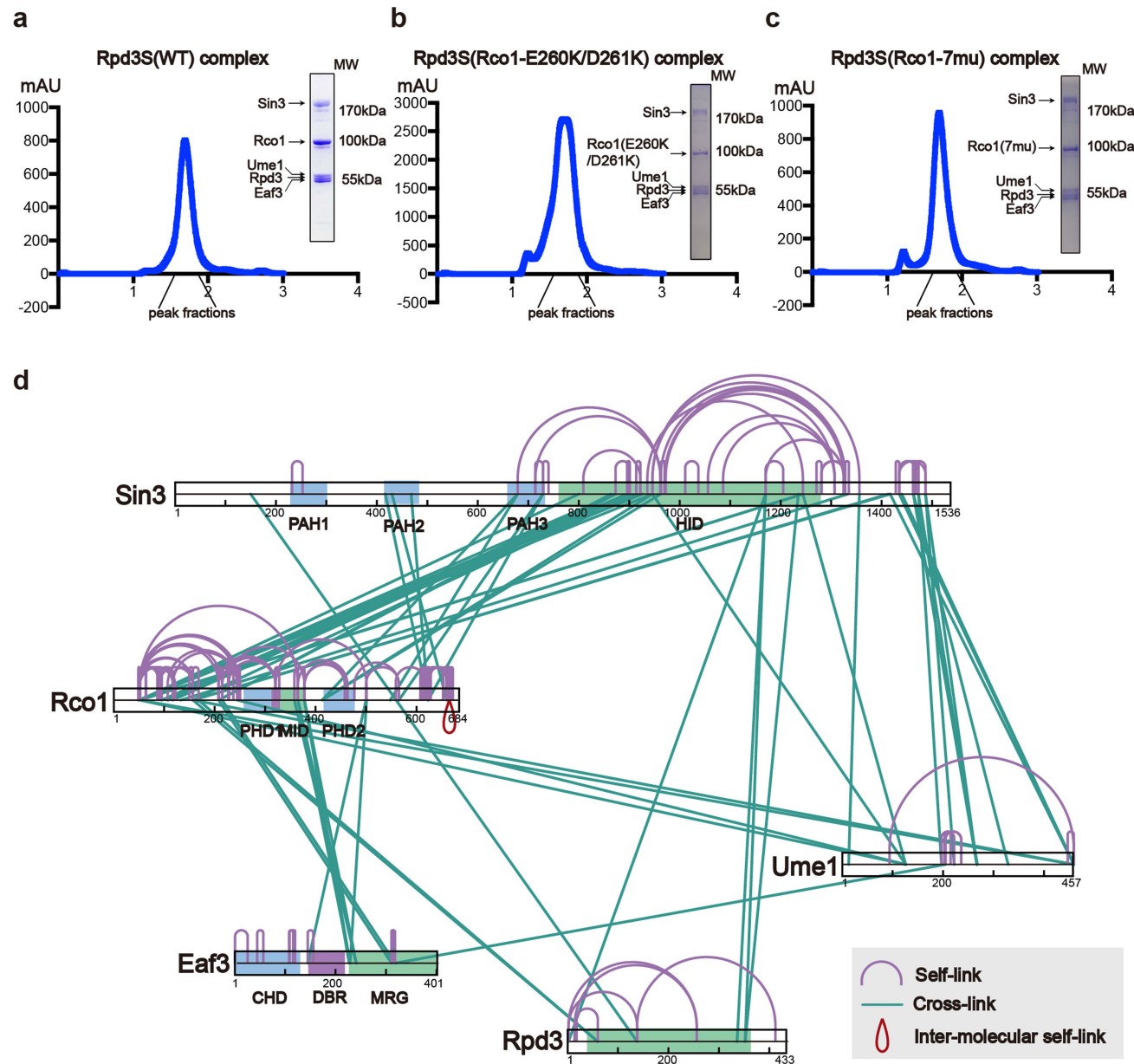

**Extended Data Fig. 1 | Protein purification and XL-MS analysis of Rpd3S complex. a-c.** Purification of Rpd3S and mutant complexes. Purification of the Rpd3S and mutant complexes. The complexes were purified using size-exclusion chromatography (Superose 6), and the peak fractions were subjected to SDS-PAGE for Coomassie blue staining. The complexes include: (**a**) wild-type of Rpd3S complex; (**b**) Rco1-E260A/D261A of the Rpd3S complex to disrupt the

PHD1 of Rco1; (**c**) Rco1-7mutants of Rpd3S complex to disrupt the left arm region. **d**. Schematic representation of the intermolecular cross-links within the Rpd3S complex. The domains of Rpd3S are indicated, and the identified inter-subunit cross-links or subunit self-links are shown as cyan or modena solid lines, respectively. The special intermolecular self-links in the C-terminal of Rco1 are shown as red solid lines.

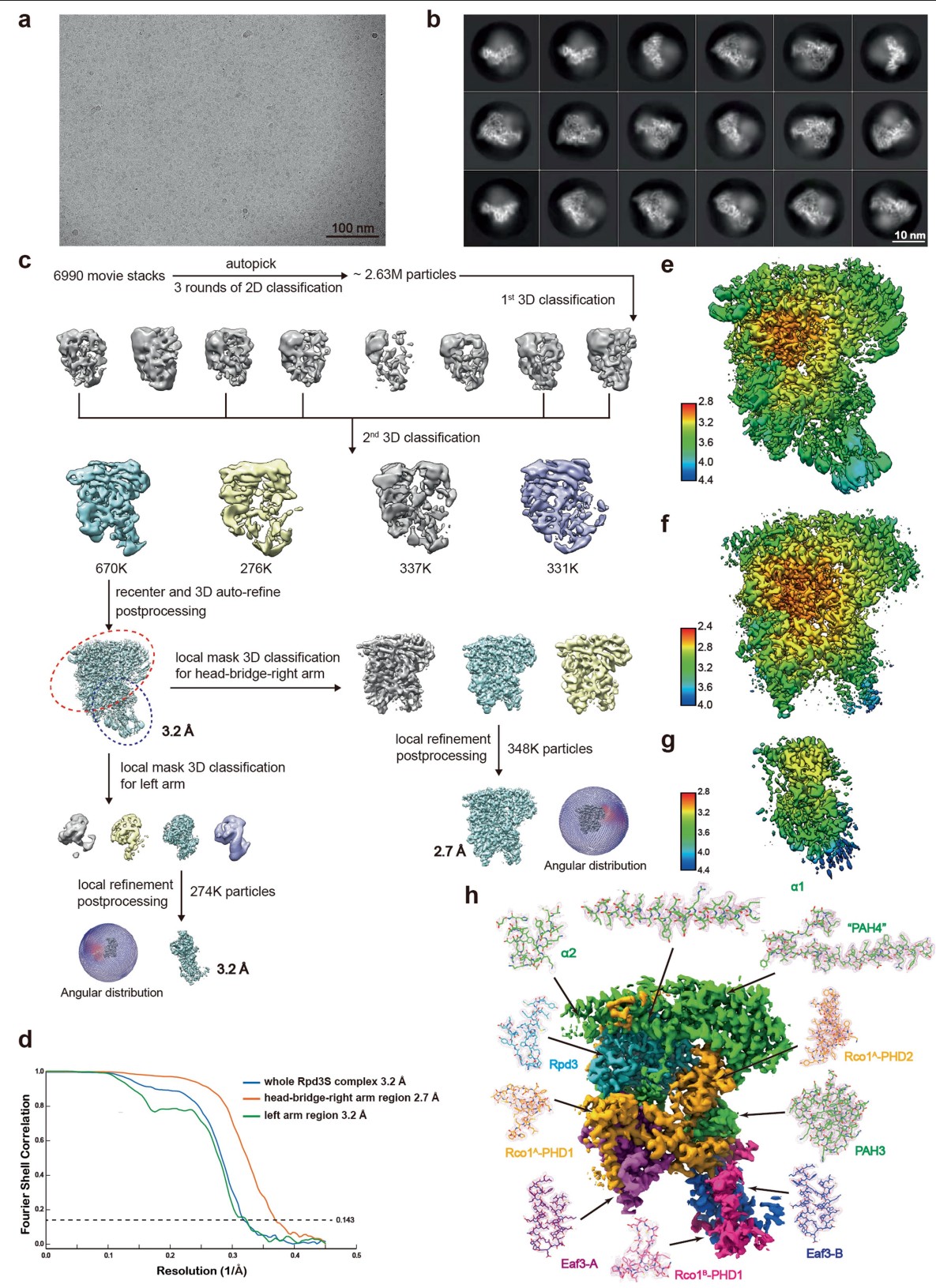

**Extended Data Fig. 2 | Data collection, image processing, Cryo-EM reconstructions and structural model of the Rpd3S complex.** Representative cryo-EM micrograph (**a**) and 2D class averages (**b**) of various projection views of the Rpd3S complex. **c**. Flowchart of the cryo-EM image processing, 3D reconstructions for the Rpd3S complex and angular distribution of EM maps. **d**. Resolution estimation of the EM maps. **e**. Estimated resolution of the cryo-EM reconstructions of the whole Rpd3S complex. Local resolution estimation of the cryo-EM reconstructions of the head-bridge-right arm region (**f**) and the bridge-left arm region (**g**) of the Rpd3S complex. **h**. The locally refined cryo-EM map of the Rpd3S complex. Close-up views of fragments of Rpd3S subunits with cryo-EM densities shown as meshes. The residues are shown as sticks representations.

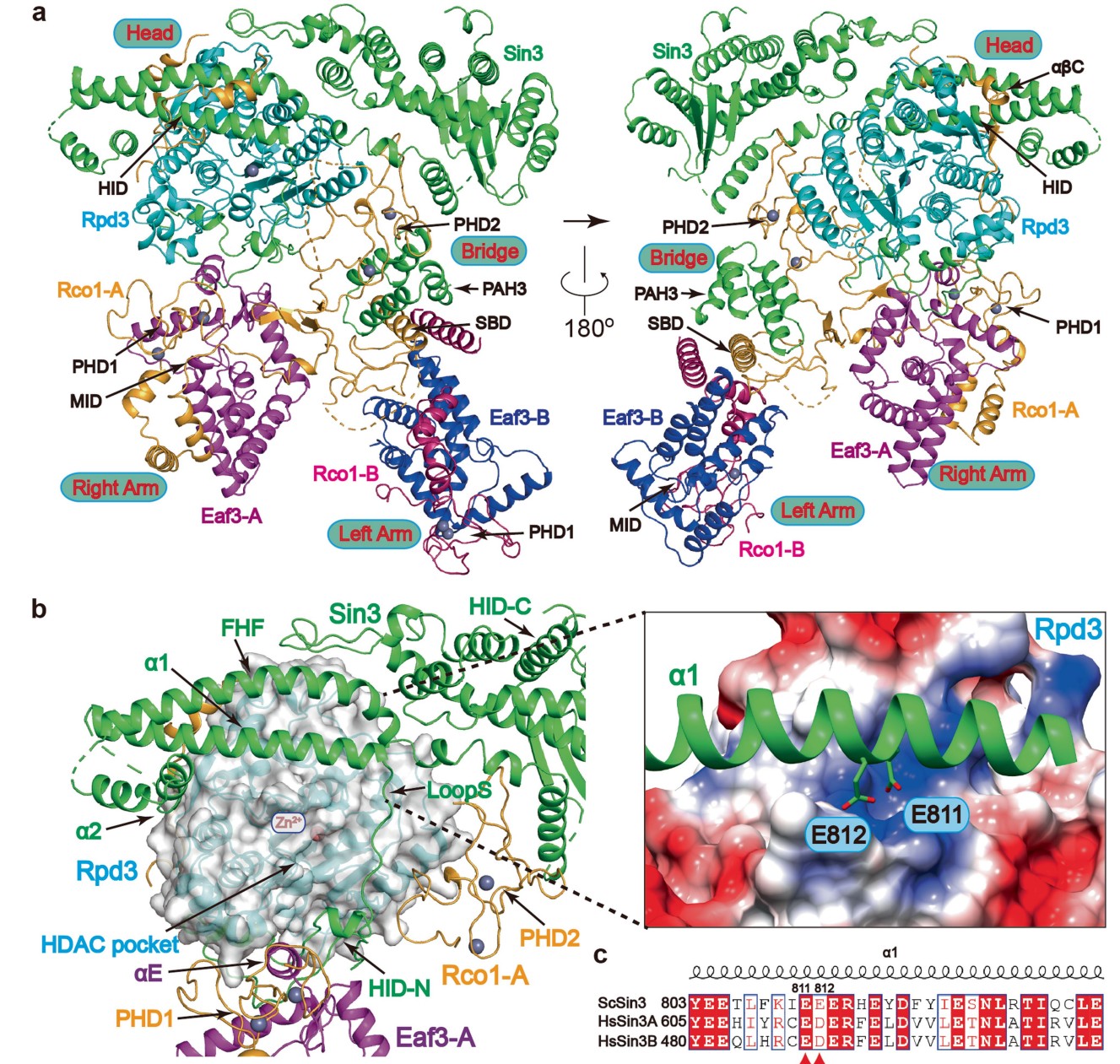

**Extended Data Fig. 3 | Detailed structure of the Rpd3S complex. a**. A cartoon model of Rpd3S complex shown in two different views. PAH, paired amphipathic helice; HID, HDAC-interaction domain; αβC, "α-β-coil" motif; PHD, plant homeobox domain; MID, MRG-interacting domain; SBD, Sin3-binding domain.

**b**. The interactions between the core enzyme Rpd3 and its neighboring subunits are shown. Acidic residues of the α1 of Sin3 pack against the basic surface of the Rpd3. The negatively charged residues are shown as sticks. **c**. Sequence conservation analysis of the α1 of Sin3 from yeast to human.

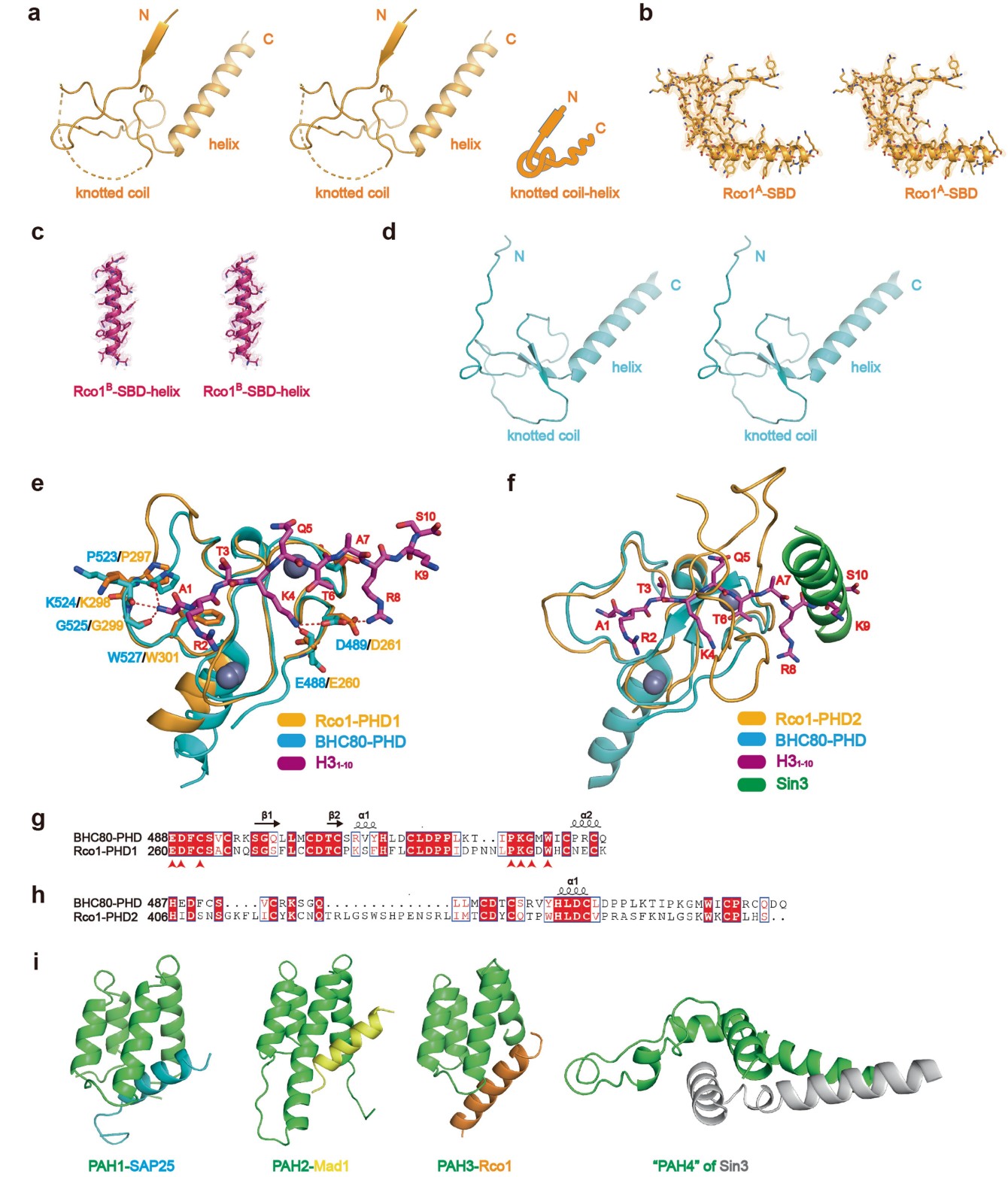

**Extended Data Fig. 4 | Detailed structure of Rco1-SBD, Rco1-PHDs and Sin3-PAH4. a**. Stereo view of the SBD domain of Rco1-A shown in cartoon form. Detailed structures of visible SBD domain of Rco1-A (**b**) and Rco1-B (**c**). Residues are depicted as sticks. **d**. Stereo view of the SBD domain of Rco1-A predicted by AlphaFold. **e** and **f**. Comparison of the PHD1 and PHD2 of Rco1 with BHC80-PHD (PDB ID: 2PUY)[61]. **g** and **h**. Sequence conservation analysis of PHD1 and PHD2 of Rco1 with BHC80-PHD. **i**. Structural comparison of PAH domains. PAH1-SAP25 (PDB ID: 2RMS)[25], PAH2-Mad1 (PDB ID: 1G1E)[26], PAH3-Rco1 in Rpd3S complex, "PAH4" of Sin3 in the Rpd3S complex. The PAH domains are colored in green.

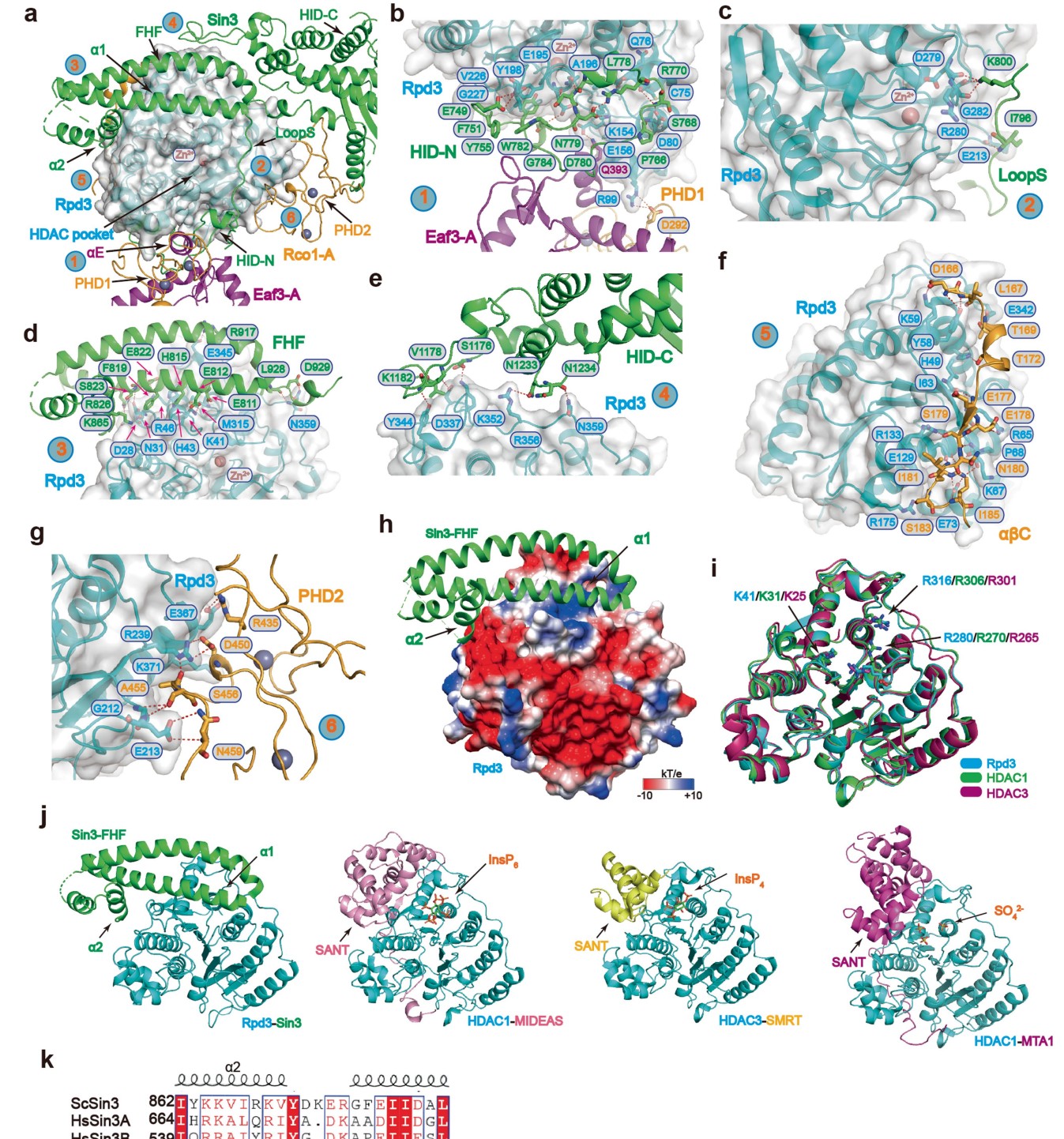

**Extended Data Fig. 5 | The interface details and regulatory modes between the core enzyme Rpd3 and its neighboring subunits. a**. A global view of the interactions around the core enzyme Rpd3. The positions of interaction are marked with numbers. Detailed views of the interactions between Rpd3 and HID-N of Sin3, PHD1 of Rco1-A, Eaf3-A (**b**); LoopS of Sin3 (**c**); FHF of Sin3 (**d**); HID-C of Sin3 (**e**); αβC of Rco1-A (**f**); PHD2 of Rco1-A (**g**). Residues at the interface are depicted as sticks. **h**. Close-up view of interactions between the HID domain of Sin3 shown in cartoon and Rpd3 shown in surface representation.

**i**. Comparison of the overall structure and key amino acids in the basic pocket of HDACs. Rpd3S is colored in blue, HDAC1 is colored in green, and HDAC3 is colored in purple. **j**. Structural comparison of HDAC complexes in inositol phosphates regulation. Rpd3S complex: Rpd3-Sin3, MiDAC complex: HDAC1-MIDEAS (PDB ID: 6Z2J)[62], SMRT complex: HDAC3-SMRT (PDB ID: 4A69)[29], NuRD complex: HDAC1-MTA1 (PDB ID: 4BKX)[28]. **k**. Sequence conservation analysis of the α2 of Sin3 from yeast to human.

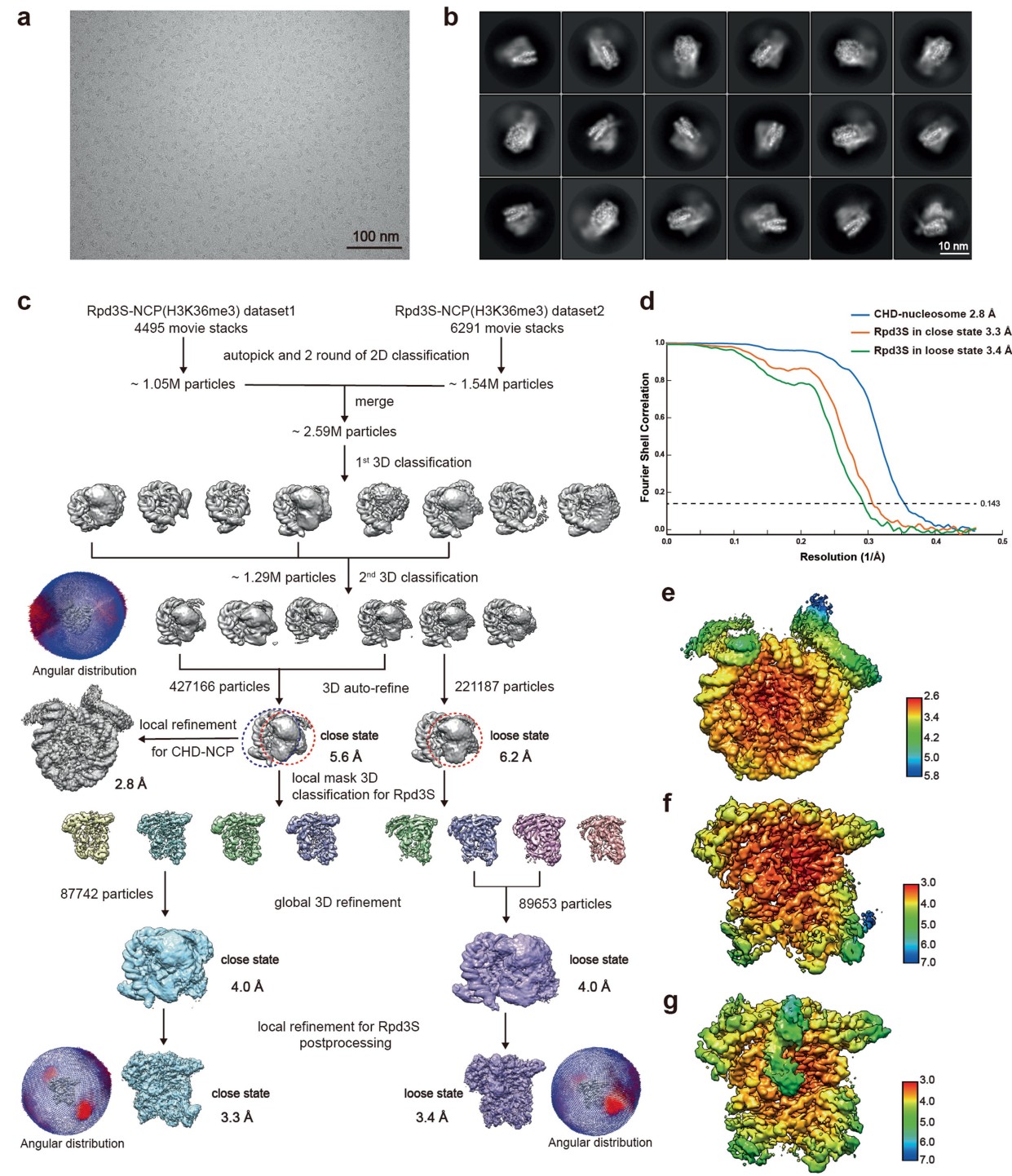

**Extended Data Fig. 6 | Data collection, image processing, Cryo-EM reconstructions and structural models of the Rpd3S-nucleosome.** Representative cryo-EM micrograph (**a**) and 2D class averages (**b**) of various projection views of Rpd3S-nucleosome. **c**. Flowcharts of the cryo-EM image processing, 3D reconstructions for the Rpd3S-nucleosome and angular distribution of EM maps. **d**. Resolution estimation of the EM maps. Local estimated resolution of the cryo-EM reconstructions of the CHD-nucleosome (**e**), Rpd3S in close state (**f**), and Rpd3S in loose state (**g**).

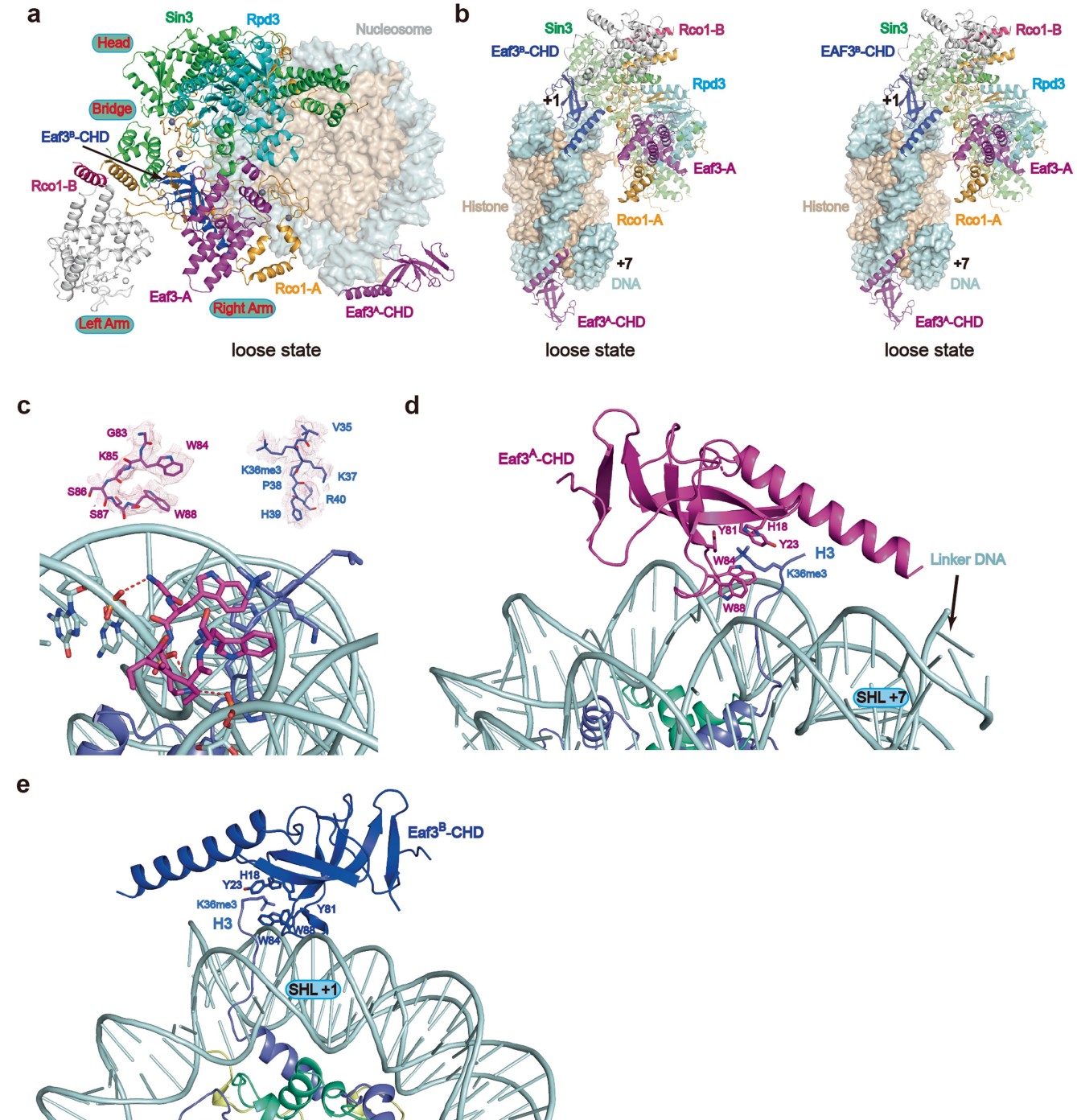

**Extended Data Fig. 7 | Cryo-EM structure of Rpd3S complex bound to H3K36me3 nucleosome in the loose state and the interface details between CHD domains and histone H3K36me3. a**. Core Rpd3S complex bound to the H3K36me3 modified nucleosome in the loose state. **b**. Stereo view of contact between core Rpd3S complex and H3K36me3 modified nucleosome in the loose state. Sin3, Rpd3, Eaf3, and Rco1 are shown in cartoon form. Nucleosome is shown in surface representation. The invisible left arm region is colored white.

**c**. Detailed view of interactions between CHD and H3 tail. Close-up views of the loop of CHD and H3 tail for interactions with cryo-EM densities shown as meshes. **d**. Detailed view of the interactions between Eaf3A-CHD and the H3K36me3 modified nucleosome. **e**. Detailed view of the interactions between Eaf3B-CHD and the H3K36me3 modified nucleosome. The residues of CHDs and H3 tail residues are shown as sticks. The positions of nucleosomal DNA are labeled with SHL numbers.

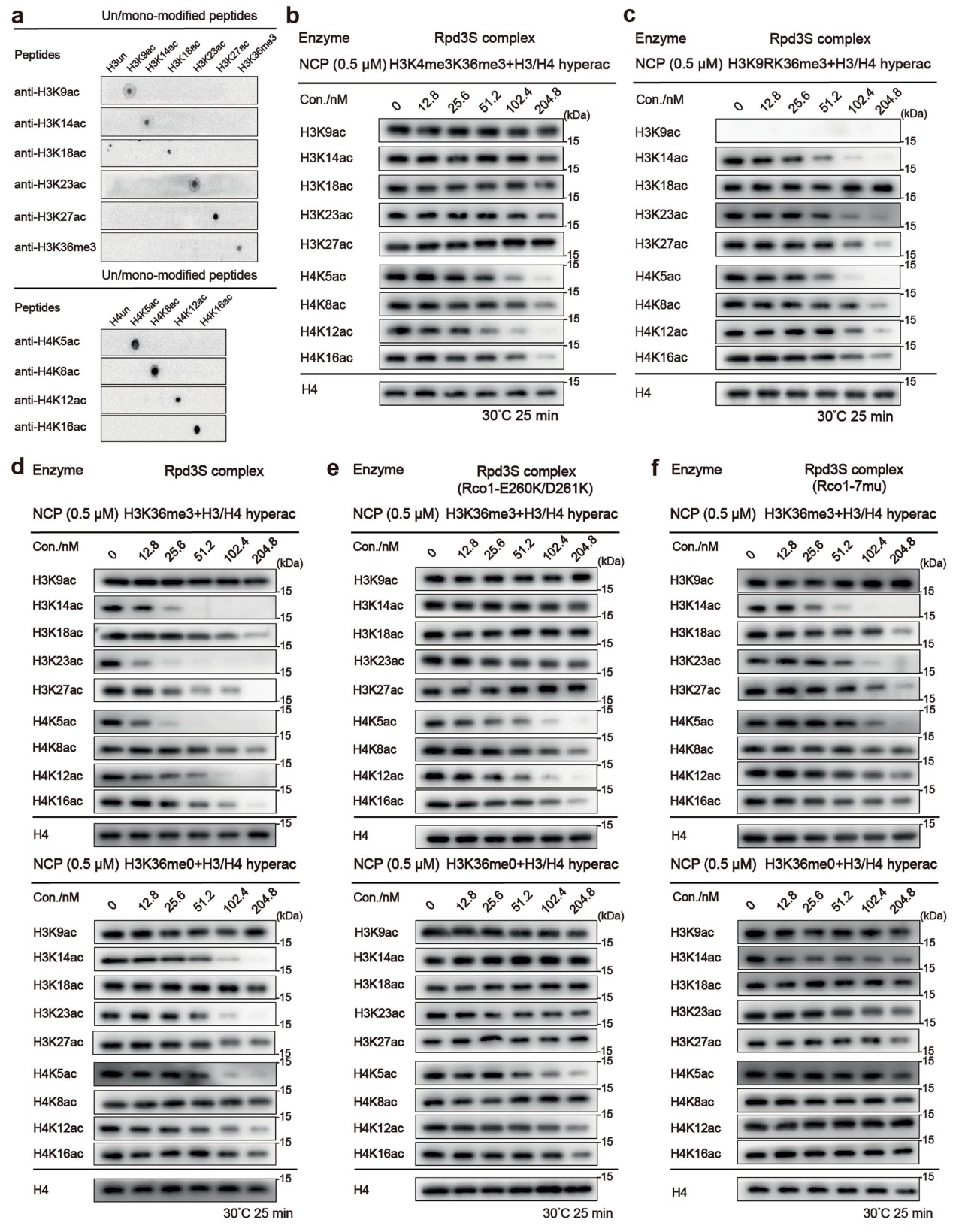

**Extended Data Fig. 8 | The catalytic activity of Rpd3S on modified nucleosomes. a.** Validation of the site-specificity of antibodies used in HDAC assay. **b.** A representative HDAC assay measuring activity of Rpd3S complex on H3K4me3K36me3 nucleosome. **c.** A representative HDAC assay measuring activity of Rpd3S complex on H3K9RK36me3 nucleosome. A representative HDAC assay measuring the activity of Rpd3S complex containing wild-type (**d**), mutants of PHD1 (**e**), and left arm region (**f**) on H3K36me3 and H3K36me0 nucleosomes. The reaction products were identified using Western blot. One representative example of three (**a**-**f**) independent experiments is shown.

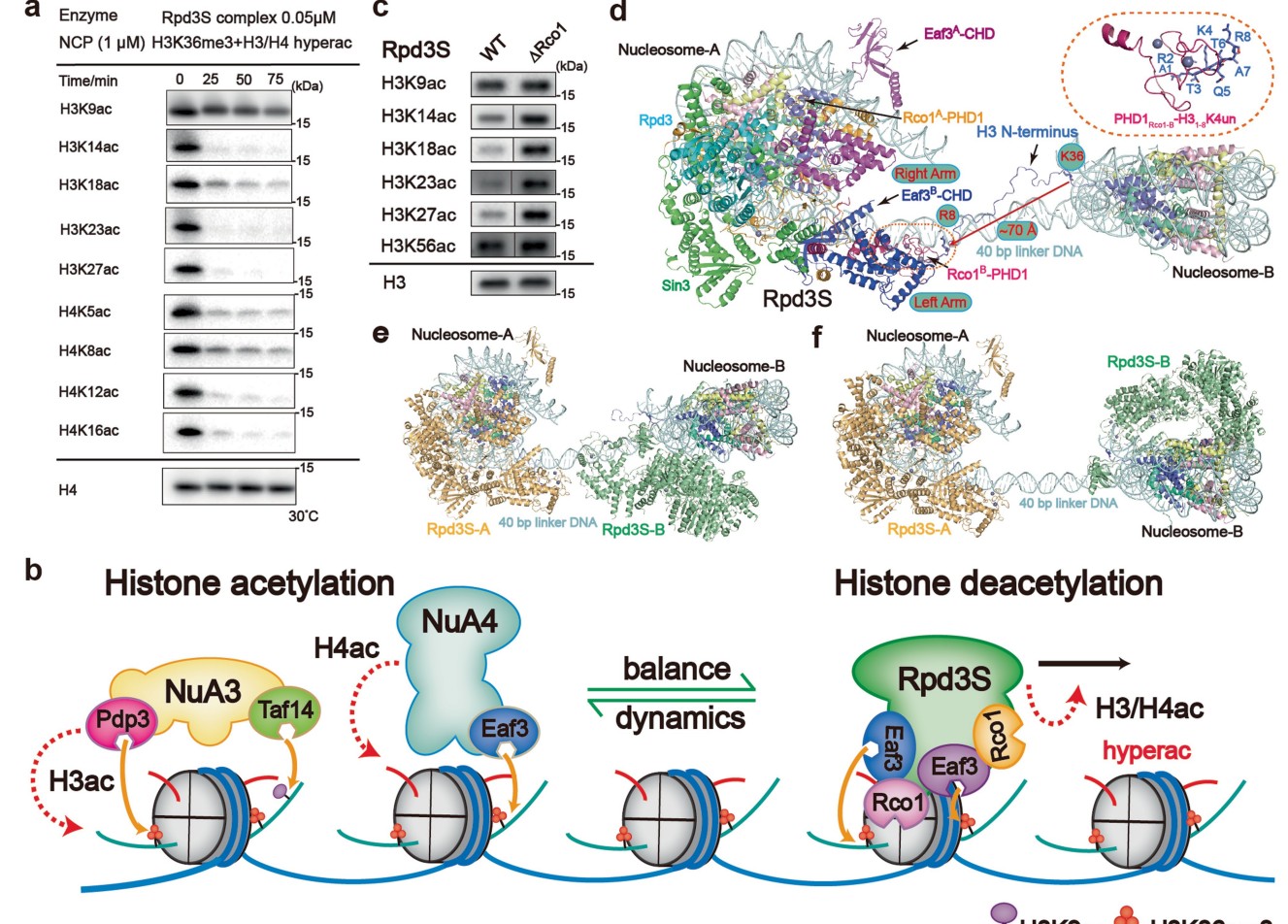

**Extended Data Fig. 9 | The regulatory models of Rpd3S complex.**
**a**. A representative HDAC assay measuring the activity of Rpd3S complex in sufficient time, where only H3K9ac can be retained over time. **b**. A "seeding mark" model of "Rpd3S-NuA3/NuA4" enzymatic pairs in balancing chromatin acetylation levels during transcription. The complexes are shown in a cartoon model. The Rpd3S complex recognizes H3K36me3 and removes most N-terminal acetylation marks of H3 and H4, except for H3K9ac. The Rpd3S-resistent H3K9ac, along with H3K36me3, may serve as "seeding marks" that can recruit NuA3 and NuA4 for the reestablishment of hyperacetylated histones H3 and H4, respectively. **c**. The H3K56ac modification is not directly affected by the Rpd3S complex *in vivo*. Western blot shows H3 acetylation levels at different sites in Rpd3S wild-type and Rco1-deleted strains. The alterations observed in H3K9ac and H3K56ac are not significant in comparison to other H3 sites in Rco1-deleted strains. The black vertical lines in the figure indicate that the rearranged lanes are from non-adjacent lanes within the same gels. The catalytic models of Rpd3S with di-nucleosome (**d**–**f**). **d**. The two CHD domains of Eaf3-A and Eaf3-B with PHD1 of Rco1-A are involved in recognizing one nucleosome, and PHD1 of Rco1-B is involved in recognizing another nucleosome. **e** and **f**. Two Rpd3S can bind to two nucleosomes, respectively, at a suitable 40bp linker DNA length. The latter Rpd3S complexes may be assembled on two nucleosome discs respectively. One representative example of three (**a**,**c**) independent experiments is shown.

**Extended Data Table 1 | Cryo-EM data collection, refinement, and validation statistics of Rpd3S complex**

|  | Rpd3S complex | Rpd3S complex (head-bridge-right arm) | Rpd3S complex (left arm) |
|---|---|---|---|
| EMDB | 33845 | 33846 | 33847 |
| PDB | 7YI0 |  |  |
| **Data collection and processing** |  |  |  |
| Magnification | 64000 | 64000 | 64000 |
| Voltage (kV) | 300 | 300 | 300 |
| Electron exposure (e-/ Å2) | 50 | 50 | 50 |
| Number of frames per movie | 32 | 32 | 32 |
| Energy filter slit width (eV) | 20 | 20 | 20 |
| Automation software | AutoEMation2 | AutoEMation2 | AutoEMation2 |
| Defocus range (µm) | -1.8 to -2.5 | -1.8 to -2.5 | -1.8 to -2.5 |
| Pixel size (Å) | 1.10 | 1.10 | 1.10 |
| Symmetry imposed | C1 | C1 | C1 |
| Micrographs (no.) | 6990 | 6990 | 6990 |
| Initial particles images (no.) | 2.63 M | 2.63 M | 2.63 M |
| Final particles images (no.) | 466 K | 466 K | 466 K |
| Map resolution (Å) | 3.2 | 2.7 | 3.2 |
| FSC threshold | 0.143 | 0.143 | 0.143 |
| Map resolution range (Å) | 2.5-3.5 | 2.5-3.5 | 2.7-3.5 |
| Map sharpening B-factor (Å) | -147.8 | -70.9 | -108.1 |
| **Refinement** |  |  |  |
| Refinement package | Phenix |  |  |
| **R.m.s. deviations** |  |  |  |
| Bond lengths (Å) | 0.003 |  |  |
| Bond angles (°) | 0.567 |  |  |
| **Validation** |  |  |  |
| MolProbity score | 1.84 |  |  |
| Clashscore | 8.46 |  |  |
| Rotamer outliers (%) | 0.93 |  |  |
| Cβ outliers (%) | 0.00 |  |  |
| CaBLAM outliers (%) | 3.31 |  |  |
| EMRinger score | 3.23 |  |  |
| **Overall correlation coefficients** |  |  |  |
| CC (mask) | 0.80 |  |  |
| CC (box) | 0.79 |  |  |
| CC (peaks) | 0.75 |  |  |
| CC (volume) | 0.81 |  |  |
| **Ramachandran plot** |  |  |  |
| Favored (%) | 94.33 |  |  |
| Allowed (%) | 5.67 |  |  |
| Disallowed (%) | 0.00 |  |  |

**Extended Data Table 2 | Cryo-EM data collection, refinement, and validation statistics of Rpd3S-nucleosome**

| | CHD-NCP | Rpd3S (close state) | Rpd3S (loose state) | Rpd3S-NCP (close state) | Rpd3S-NCP loose state) |
|---|---|---|---|---|---|
| EMDB | 33848 | 33850 | 33849 | 33851 | 33852 |
| PDB | 7YI1 | 7YI3 | 7YI2 | 7YI4 | 7YI5 |
| **Data collection and processing** | | | | | |
| Magnification | 64000 | 64000 | 64000 | 64000 | 64000 |
| Voltage (kV) | 300 | 300 | 300 | 300 | 300 |
| Electron exposure (e-/ Å2) | 50 | 50 | 50 | 50 | 50 |
| Number of frames per movie | 32 | 32 | 32 | 32 | 32 |
| Energy filter slit width (eV) | 20 | 20 | 20 | 20 | 20 |
| Automation software | AutoEMation2 | AutoEMation2 | AutoEMation2 | AutoEMation2 | AutoEMation2 |
| Defocus range (μm) | -1.8 to -2.5 | -1.8 to -2.5 | -1.8 to -2.5 | -1.8 to -2.5 | -1.8 to -2.5 |
| Pixel size (Å) | 1.08 | 1.08 | 1.08 | 1.08 | 1.08 |
| Symmetry imposed | C1 | C1 | C1 | C1 | C1 |
| Micrographs (no.) | 10786 | 10786 | 10786 | 10786 | 10786 |
| Initial particles images (no.) | 2.59 M | 2.59 M | 2.59 M | 2.59 M | 2.59 M |
| Final particles images (no.) | 427 K | 87.7 K | 89.6 K | 87.7 K | 89.6 K |
| Map resolution (Å) | 2.8 | 3.3 | 3.4 | 4.0 | 4.0 |
| FSC threshold | 0.143 | 0.143 | 0.143 | 0.143 | 0.143 |
| Map resolution range (Å) | 2.5-6.0 | 2.8-5.8 | 2.8-6.0 | 3.5-9.5 | 3.5-10.0 |
| Map sharpening B-factor (Å) | -85.2 | -85.0 | -100.5 | -113.143 | -120.048 |
| **Refinement** | | | | | |
| Refinement package | Phenix | Phenix | Phenix | Phenix | Phenix |
| **R.m.s. deviations** | | | | | |
| Bond lengths (Å) | 0.003 | 0.003 | 0.003 | 0.004 | 0.005 |
| Bond angles (°) | 0.562 | 0.523 | 0.594 | 0.608 | 0.662 |
| **Validation** | | | | | |
| MolProbity score | 1.40 | 1.83 | 1.95 | 1.81 | 2.01 |
| Clashscore | 7.33 | 7.77 | 9.63 | 10.64 | 15.59 |
| Rotamer outliers (%) | 0.37 | 0.00 | 0.23 | 0.00 | 0.00 |
| Cβ outliers (%) | 0.00 | 0.00 | 0.00 | 0.00 | 0.00 |
| CaBLAM outliers (%) | 0.98 | 2.93 | 3.59 | 2.18 | 2.35 |
| EMRinger score | 2.94 | 2.60 | 2.10 | 0.23 | 0.30 |
| **Overall correlation coefficients** | | | | | |
| CC (mask) | 0.84 | 0.77 | 0.77 | 0.82 | 0.84 |
| CC (box) | 0.81 | 0.73 | 0.75 | 0.85 | 0.87 |
| CC (peaks) | 0.74 | 0.62 | 0.62 | 0.77 | 0.78 |
| CC (volume) | 0.82 | 0.75 | 0.75 | 0.82 | 0.84 |
| **Ramachandran plot** | | | | | |
| Favored (%) | 98.09 | 93.91 | 93.12 | 96.13 | 95.49 |
| Allowed (%) | 1.91 | 6.09 | 6.88 | 3.87 | 4.51 |
| Disallowed (%) | 0.00 | 0.00 | 0.00 | 0.00 | 0.00 |

# Reporting Summary

## Statistics

For all statistical analyses, confirm that the following items are present in the figure legend, table legend, main text, or Methods section.

| n/a | Confirmed | |
|---|---|---|
| ☐ | ☒ | The exact sample size (*n*) for each experimental group/condition, given as a discrete number and unit of measurement |
| ☐ | ☒ | A statement on whether measurements were taken from distinct samples or whether the same sample was measured repeatedly |
| ☒ | ☐ | The statistical test(s) used AND whether they are one- or two-sided *Only common tests should be described solely by name; describe more complex techniques in the Methods section.* |
| ☒ | ☐ | A description of all covariates tested |
| ☒ | ☐ | A description of any assumptions or corrections, such as tests of normality and adjustment for multiple comparisons |
| ☐ | ☒ | A full description of the statistical parameters including central tendency (e.g. means) or other basic estimates (e.g. regression coefficient) AND variation (e.g. standard deviation) or associated estimates of uncertainty (e.g. confidence intervals) |
| ☒ | ☐ | For null hypothesis testing, the test statistic (e.g. *F*, *t*, *r*) with confidence intervals, effect sizes, degrees of freedom and *P* value noted *Give P values as exact values whenever suitable.* |
| ☒ | ☐ | For Bayesian analysis, information on the choice of priors and Markov chain Monte Carlo settings |
| ☒ | ☐ | For hierarchical and complex designs, identification of the appropriate level for tests and full reporting of outcomes |
| ☒ | ☐ | Estimates of effect sizes (e.g. Cohen's *d*, Pearson's *r*), indicating how they were calculated |

*Our web collection on statistics for biologists contains articles on many of the points above.*

## Software and code

Policy information about availability of computer code

| Data collection | We used AutoEMation (version2.0) to collect all cryo-EM datasets, written by Dr. Jianlin Lei at Tsinghua University. |
|---|---|
| Data analysis | MotionCor2, GCTF v1.06, RELION 3.1.3, UCSF Chimera 1.16, UCSF Chimera X 1.2.5, COOT 0.9, Phenix-1.15.2-3472, Pymol 2.3.2, MolProbity 4.2 |

For manuscripts utilizing custom algorithms or software that are central to the research but not yet described in published literature, software must be made available to editors and reviewers. We strongly encourage code deposition in a community repository (e.g. GitHub). See the Nature Portfolio guidelines for submitting code & software for further information.

## Data

Policy information about availability of data

All manuscripts must include a data availability statement. This statement should provide the following information, where applicable:
- Accession codes, unique identifiers, or web links for publicly available datasets
- A description of any restrictions on data availability
- For clinical datasets or third party data, please ensure that the statement adheres to our policy

The authors declare that the data supporting the findings of this study are available within the paper. The cryo-EM density maps have been deposited in the Electron Microscopy Data Bank under EMD accession codes 33845, 33846, 33847, 33848, 33849, 33850, 33851, 33852. The coordinates of the atomic model have been deposited in the Protein Data Bank under ID codes 7YI0, 7YI1, 7YI2, 7YI3, 7YI4, 7YI5. Several structural coordinates in the PDB database were used in this study, which can be located by accession numbers 6ESF and 3E9G. Source Data is available online.

# Research involving human participants, their data, or biological material

Policy information about studies with human participants or human data. See also policy information about sex, gender (identity/presentation), and sexual orientation and race, ethnicity and racism.

| Reporting on sex and gender | *Use the terms sex (biological attribute) and gender (shaped by social and cultural circumstances) carefully in order to avoid confusing both terms. Indicate if findings apply to only one sex or gender; describe whether sex and gender were considered in study design; whether sex and/or gender was determined based on self-reporting or assigned and methods used.*<br>*Provide in the source data disaggregated sex and gender data, where this information has been collected, and if consent has been obtained for sharing of individual-level data; provide overall numbers in this Reporting Summary. Please state if this information has not been collected.*<br>*Report sex- and gender-based analyses where performed, justify reasons for lack of sex- and gender-based analysis.* |
|---|---|
| Reporting on race, ethnicity, or other socially relevant groupings | *Please specify the socially constructed or socially relevant categorization variable(s) used in your manuscript and explain why they were used. Please note that such variables should not be used as proxies for other socially constructed/relevant variables (for example, race/ethnicity should not be used as a proxy for socioeconomic status).*<br>*Provide clear definitions of the relevant terms used, how they were provided (by the participants/respondents, the researchers, or third parties), and the method(s) used to classify people into the different categories (e.g. self-report, census or administrative data, social media data, etc.)*<br>*Please provide details about how you controlled for confounding variables in your analyses.* |
| Population characteristics | *Describe the covariate-relevant population characteristics of the human research participants (e.g. age, genotypic information, past and current diagnosis and treatment categories). If you filled out the behavioural & social sciences study design questions and have nothing to add here, write "See above."* |
| Recruitment | *Describe how participants were recruited. Outline any potential self-selection bias or other biases that may be present and how these are likely to impact results.* |
| Ethics oversight | *Identify the organization(s) that approved the study protocol.* |

Note that full information on the approval of the study protocol must also be provided in the manuscript.

# Field-specific reporting

Please select the one below that is the best fit for your research. If you are not sure, read the appropriate sections before making your selection.

☒ Life sciences          ☐ Behavioural & social sciences          ☐ Ecological, evolutionary & environmental sciences

For a reference copy of the document with all sections, see nature.com/documents/nr-reporting-summary-flat.pdf

# Life sciences study design

All studies must disclose on these points even when the disclosure is negative.

| Sample size | Sample sizes were not pre-determined. Cryo-EM images were collected until structures of satisfactory quality were solved, which suggested sufficient sample size. For biochemical assays, we performed two to three replicates since the replications are successful. |
|---|---|
| Data exclusions | No data were excluded during structural analysis. |
| Replication | Our HDAC assays, spotting assays and western blotting were performed in two to three independent replicates, all attempts at replication were successful. No data was excluded. |
| Randomization | No group allocation was performed in structural experiments. |
| Blinding | Blinding was not performed as subjective analysis was not needed and no group allocation was performed for structural experiments. |

# Reporting for specific materials, systems and methods

We require information from authors about some types of materials, experimental systems and methods used in many studies. Here, indicate whether each material, system or method listed is relevant to your study. If you are not sure if a list item applies to your research, read the appropriate section before selecting a response.

## Materials & experimental systems

| n/a | Involved in the study |
|---|---|
| ☐ | ☒ Antibodies |
| ☐ | ☒ Eukaryotic cell lines |
| ☒ | ☐ Palaeontology and archaeology |
| ☒ | ☐ Animals and other organisms |
| ☒ | ☐ Clinical data |
| ☒ | ☐ Dual use research of concern |
| ☒ | ☐ Plants |

## Methods

| n/a | Involved in the study |
|---|---|
| ☒ | ☐ ChIP-seq |
| ☒ | ☐ Flow cytometry |
| ☒ | ☐ MRI-based neuroimaging |

# Antibodies

| | |
|---|---|
| Antibodies used | Anti-Acetyl-Histone H3 (Lys14) Mouse mAb (PTM BIO, Cat.#: PTM-157, Dilution: 1:500), Anti-Acetyl-Histone H3 (Lys18) Mouse mAb (PTM BIO, Cat.#: PTM- 158, Dilution: 1:750), Anti-Acetyl-Histone H3 (Lys23) Rabbit pAb(PTM BIO, Cat.#: PTM- 115, Dilution: 1:1500), Anti-Acetyl-Histone H3 (Lys27) Mouse mAb (PTM BIO, Cat.#: PTM- 160, Dilution: 1:2000), Anti-Tri-Methyl-Histone H3 (Lys36) Rabbit pAb (PTM BIO, Cat.#: PTM- 625, Dilution: 1:1000), Anti-Histone H4 Mouse mAb (NT) (PTM BIO, Cat.#: PTM- 1009, Dilution: 1:1000), Histone H3 Rabbit pAb(ABclonal, Cat.#: A2348, Dilution: 1:2000), Anti-Acetyl-Histone H3 (Lys56) Mouse mAb (PTM BIO, Cat.#: PTM- 162, Dilution: 1:1000), Acetyl-Histone H3-K9 Rabbit pAb(ABclonal, Cat.#: A7255, Dilution: 1:5000), Acetyl-Histone H4-K8 Rabbit pAb (ABclonal, Cat.#: A7258, Dilution: 1:6000), Acetyl-Histone H4-K12 Rabbit mAb(ABclonal, Cat.#: A22754, Dilution: 1:1000), Anti-acetyl-Histone H4 (Lys5) Antibody, rabbit monoclonal(Sigma-Aldrich, Cat.#: 04-118, Dilution:1:5000), and Acetyl-Histone H4 (Lys16) (E2B8W) Rabbit mAb((Cell Signaling Technology, Cat.#: 13534S, Dilution:1:500). |
| Validation | The antibodies employed in this research were commercially obtainable and were authenticated by the provider based on the information presented in the relevant data sheets.<br>Anti-Acetyl-Histone H3 (Lys14) Mouse mAb (PTM BIO, Cat.#: PTM-157): https://www.ptmbiolabs.com/product/ptm-157/<br>Anti-Acetyl-Histone H3 (Lys18) Mouse mAb (PTM BIO, Cat.#: PTM- 158): https://www.ptmbiolabs.com/product/ptm-158/<br>Anti-Acetyl-Histone H3 (Lys23) Rabbit pAb (PTM BIO, Cat.#: PTM- 115): https://www.ptmbiolabs.com/product/ptm-115/<br>Anti-Acetyl-Histone H3 (Lys27) Mouse mAb (PTM BIO, Cat.#: PTM- 160): https://www.ptmbiolabs.com/product/ptm-160/<br>Anti-Tri-Methyl-Histone H3 (Lys36) Rabbit pAb (PTM BIO, Cat.#: PTM- 625): https://www.ptmbiolabs.com/product/ptm-625/<br>Anti-Histone H4 Mouse mAb (NT) (PTM BIO, Cat.#: PTM- 1009): https://www.ptmbiolabs.com/product/pan-h4-mouse-mab/<br>H3 Rabbit pAb(ABclonal, Cat.#: A2348): https://abclonal.com/catalog-antibodies/HistoneH3RabbitPAb/A2348<br>Anti-Acetyl-Histone H3 (Lys56) Mouse mAb (PTM BIO, Cat.#: PTM- 162): https://www.ptmbiolabs.com/product/ptm-162/<br>Acetyl-Histone H3-K9 Rabbit pAb(ABclonal, Cat.#: A7255): https://abclonal.com/catalog-antibodies/AcetylHistoneH3K9RabbitpAb/A7255<br>Acetyl-Histone H4-K8 Rabbit pAb (ABclonal, Cat.#: A7258): https://abclonal.com/catalog-antibodies/AcetylHistoneH4K8RabbitpAb/A7258<br>Acetyl-Histone H4-K12 Rabbit mAb(ABclonal, Cat.#: A22754): https://abclonal.com/catalog-antibodies/AcetylHistoneH4K12RabbitmAb/A22754<br>Anti-acetyl-Histone H4 (Lys5) Antibody, rabbit monoclonal(Sigma-Aldrich, Cat.#: 04-118): https://www.sigmaaldrich.cn/CN/en/product/mm/04118<br>Acetyl-Histone H4 (Lys16) (E2B8W) Rabbit mAb((Cell Signaling Technology, Cat.#: 13534S): https://www.cellsignal.com/products/primary-antibodies/acetyl-histone-h4-lys16-e2b8w-rabbit-mab/13534?_=1682335024811&Ntt=13534S&tahead=true |

# Eukaryotic cell lines

Policy information about cell lines and Sex and Gender in Research

| | |
|---|---|
| Cell line source(s) | SF9, obtained from invitrogen |
| Authentication | Cell lines were directly purchased from Invitrogen. Cell line authentication was not performed at our hand during cell culture. |
| Mycoplasma contamination | Cell lines had been tested negative for mycoplasma contamination by Invitrogen before purchase. They were not tested at our hand during cell culture. |
| Commonly misidentified lines (See ICLAC register) | No commonly misidentified cell lines were used in the study. |

