## [Peer Review File · Nature]

Manuscript Title: Diverse modes of H3K36me₃-guided nucleosomal deacetylation by Rpd3S

Reviewer Comments & Author Rebuttals

Reviewer Reports on the Initial Version:

Referees' comments:

Referee #1 (Remarks to the Author):

In this paper, the authors provide interesting new insight into the assembly and function of the Rpd3S histone deacetylase complex. Rpd3S is known for associating with transcribed regions through interaction with nucleosome that are H3K4me₀ and then binds to H3K36me₃ mediated by Set2 to deacetylate histones to prevent cryptic transcription and histone exchange. Here, the authors solve the cryo-EM structures of Rpd3S in isolation and on H3K36me₃-containing nucleosomes. The first interesting finding is that the complex has an asymmetric assembly of Eaf3-Rco1 heterodimers, and that each complex contains two copies of Eaf3 despite prior works suggesting one Eaf3 copy. The second interesting finding is that the asymmetric assembly of Eaf3-Rco1 allows a Rpd3S to engage nucleosomes such that one copy is positioned on H3K4me₀ and is in an orientation to remove H3 acetylations (but not H3K9), whereas the other orientation will remove H4ac. In all, the Rpd3S complex has been exquisitely evolved to deal with multivalent engagement and appropriate deacetylation of H3 and H4, but not other histones like H2A and H2B. In vivo testing of these interactions confirms the structural findings.

In all, this is an interesting story and adds to our understanding of multivalent engagement and combinatorial readout. While not an expert on cryo-EM, the study is significant and furthers our understanding of how Rpd3S functions. One weakness of the study is that Rpd3S normally uses di-nucleosomes for robust deacetylation, which has not been considered here. Is the need for di-nucleosomes due to the fact that one Rpd3S complex sits on one nucleosome to deacetylate H3 and the other Rpd3S complex sits on the di-nucleosome neighbor to remove the H4ac? And if this is the case, do two Rpd3S complexes interact or is there an opportunity for both Rpd3S complexes to "swap" positions to take care of the other H3 or H4 tail acetylation. Alternatively, maybe it is that two Rpd3S complexes can bind to a single nucleosome to remove both H3 and H4 acetylation on an individual nucleosome, but this would have been observed. Thus, a gap here is how H3 and H4 tails are simultaneously deacetylated on the known in vivo template for Rpd3S (a di-nucleosome). The significance of the work would be elevated by addition of using di-nucleosomes in the study to determine how H3 and H4 tails are deacetylated under the in vivo conditions that are known for Rpd3S. Using the correct physiological template would most likely resolve this gap. That being said, the challenges of this experiment (especially for cryo-EM) are great and it may not be technically possible at this time. But if it could be addressed, this would create the greatest impact to the field.

Finally, the inability of H3K9ac is interesting but the authors forgot to consider that Rt109 acetylates this site (and K56) to promote Asf1 histone exchange. Does this mean that Rpd3S is not involved in histone exchange? this would be interesting to examine.

Minor comment: the writing and clarity of the text needs some slight improvement.

Referee #2 (Remarks to the Author):

The Rpd3S complex is a histone deacetylase belonging to the class I HDAC family that is involved in transcriptional repression by deacetylating histones through recognition of H3K36me3. Its three-dimensional structure and the mechanism of histone deacetylation were not elucidated to date. In this study, the authors successfully resolved the three-dimensional structure of the yeast Rpd3S catalytic core complex, a giant complex of 0.6Mda on its own, and the structure of the Rpd3S complex bound to the H3K36me3 nucleosome using cryo-electron microscopy. They also generated structure-based mutants and performed various site-specific deacetylation assays, leading the authors to propose a unique mode of nucleosomal substrate recognition mechanism by the Rpd3S complex.

This paper provides the first structural description of the class I HDAC family and a well-dissected mechanistic overview of the recognition of H3K36me3 by the Rpd3S complex and its binding mode to histone H3 and H4 tails, findings that can be considered as very important for understanding the epigenetic regulation of chromatin by the class I HDAC family.

There is no doubt that this study will contribute to the development of research on the Rpd3S complex. The quality of the cryo-EM structure determined, as well as the *in vitro* and *in vivo* analyses are satisfactory. The manuscript deserves to be published, provided that the authors address the comments listed below.

Major comments:

1. The authors should include the H3K36me0 control in the HDAC assay of Rpd3S Rco1 E260K/D261K and Rco1 7mu, mutants impaired in their deacetylation activities, because we do not know whether the results obtained occur in an H3K36me3-dependent manner.
2. The authors analyzed the effects of Rco1 E260K/D261K and Rco1 7mu *in vitro* by HDAC assay, but only Rco1 E260K/D261K was analyzed *in vivo*. Therefore, the authors should investigate the effects of Rco1 7mu *in vivo* as well.
3. To investigate the importance of H3K9ac, the authors generated an H3K9R mutant and studied its effect *in vivo*. The authors state that the introduction of the H3K9R mutation reduced acetylation of histones H3 and H4 by HATs such as NuA3 and NuA4. However, it is also possible that it was enhanced deacetylation by Rpd3S. This could be easily verified by the authors by performing an HDAC assay with H3K9R *in vitro*.
4. In the introduction, the authors mention an interesting study showing that deacetylase activity may vary with the length of the linker DNA. However, the authors do not discuss this at all in the discussion. It would be very interesting if the authors could talk over this hypothesis based on their structure obtained in this study, to further the discussion.
5. In Figure 4, the angles of the overall structure and those of the close-up view are very different and are difficult to understand. The authors should correct the angles to make the figure more understandable to readers.
6. Extended Data Figure 12: it is not clear whether it is Eaf3-A CHD at SHL+7 or Eaf3-B CHD at SHL+1 that binds to H3K36me3. Also, the binding positions of the nucleosomes are different, so the authors should show independently the two nucleosomes with the different binding positions in the figure.
7. It is not clear whether it is Eaf3-A CHD that binds SHL+7 or Eaf3-B CHD that binds at SHL+1. In addition, the binding positions of the nucleosomes are different, so the authors should independently show the two nucleosomes with the different binding positions in the figure.

Minor comments

1. The following should be added to the cryo-EM structural analysis figures:
 - Extended Data Figure 2a, 9a: scale bar
 - Extended Data Figure 2b, 9b: box size
 - Extended Data Figure 2, 9: Euler angle distribution map
2. P.6 line135: "A PAH4 domain (1143-1216) has been reported to exist within Sin3." The authors need to cite a reference.
3. Extended Data Figure 10a: the nucleosome map is inverted. The authors must correct it.
4. A space is required between the number and the unit below.
Figure 5a bottom right : 25min → 25 min
Figure 6C: 100µg/ml → 100 µg/ml

Referee #3 (Remarks to the Author):

The manuscript by Guan and colleagues describes the structure of the yeast Rpd3S histone deacetylase complex in its apo and nucleosome bound forms. Rpd3S is an important histone deacetylase complex that functions in gene regulation and suppression of aberrant cryptic transcription by interacting with methylated H3K36 that is associated with transcribed genes. This work describes the structure of one of the most well-known and studied epigenetic erasers in chromatin biology, and reveals the molecular basis of recognition between these enzymatic complexes and their nucleosome substrates, and how that recognition allows tuning of the catalytic targets of the deacetylase enzyme which is explored further using structure-guided biochemical assays. HDAC-nucleosome structures are beginning to emerge in the field, but this work provides one of the most detailed and comprehensive studies to date. The work is high quality and a major advance in chromatin biology. I imagine it will be of great interest to the community and beyond, and thus suitable for publication in Nature. I only have these minor comments:

- A knotted coil is a very rare topology. How sure are the authors that this is really so, given that the supplementary figure 4 shows a significant un-modelled portion of the coil, but whose path is threaded through a loop (and forming the knot) in their schematic?
- I found the paragraph describing the two arms (lines 102-119) extremely difficult to interpret. It took a lot of effort and i'm still unsure whether i understand the authors description. One of the phrases that were the most confusing are "...the helix element of Rco1-B SBD participates in SBD surface engagement despite the invisibility of the knotted coil motif". Admittedly, the structure is very complex but i'd welcome a clearer description if possible.
- The structural basis described for crosstalk between unmodified H3K4 and H3 deacetylation is reasonable, and supported by some studies (e.g. reference 13). But it would be far more compelling if the deacetylation experiment performed in Figure 5a was applied to nucleosomes bearing H3K4me1/2/3. If their model is correct, this should reduce/prevent deacetylation in the same manner as the Rco1 mutants.

Author Rebuttals to Initial Comments:

Response Letter for “Dynamic and multivalent engagement determines context-dependent nucleosomal deacetylation by the Rpd3S complex” (nature-2022-09-14554)

Referee #1 (Remarks to the Author):

In this paper, the authors provide interesting new insight into the assembly and function of the Rpd3S histone deacetylase complex. Rpd3S is known for associating with transcribed regions through interaction with nucleosome that are H3K4me0 and then binds to H3K36me3 mediated by Set2 to deacetylate histones to prevent cryptic transcription and histone exchange. Here, the authors solve the cryo-EM structures of Rpd3S in isolation and on H3K36me3-containing nucleosomes. The first interesting finding is that the complex has an asymmetric assembly of Eaf3-Rco1 heterodimers, and that each complex contains two copies of Eaf3 despite prior works suggesting one Eaf3 copy. The second interesting finding is that the asymmetric assembly of Eaf3-Rco1 allows a Rpd3S to engage nucleosomes such that one copy is positioned on H3K4me0 and is in an orientation to remove H3 acetylations (but not H3K9), whereas the other orientation will remove H4ac. In all, the Rpd3S complex has been exquisitely evolved to deal with multivalent engagement and appropriate deacetylation of H3 and H4, but not other histones like H2A and H2B. In vivo testing of these interactions confirms the structural findings. In all, this is an interesting story and adds to our understanding of multivalent engagement and combinatorial readout. While not an expert on cryo-EM, the study is significant and furthers our understanding of how Rpd3S functions.

Response: We thank this reviewer for his/her recognition of our work by stating that “... *this is an interesting story and adds to our understanding of multivalent engagement and combinatorial readout*” and “...*the study is significant and furthers our understanding of how Rpd3S functions*”

1. Is the need for di-nucleosomes due to the fact that one Rpd3S complex sits on one nucleosome to deacetylate H3 and the other Rpd3S complex sits on the di-nucleosome neighbor to remove the H4ac? And if this is the case, do two Rpd3S complexes interact or is there an opportunity for both Rpd3S complexes to “swap” positions to take care of the other H3 or H4 tail acetylation. Alternatively, maybe it is that two Rpd3S complexes can bind to a single nucleosome to remove both H3 and H4 acetylation on an individual nucleosome, but this would have been observed.

Response: We thank the reviewer for these insightful comments and speculations. In fact, we began our structural studies using both di-nucleosome (with a linker length of 35 bp) and single nucleosome substrates. However, we were only able to solve high-resolution structures (2.8-4.0 Å) of Rpd3S with a single nucleosome. At least under the tested conditions, the use of di-nucleosome substrate was not rewarding. The resolution of the di-nucleosome data was either low or could only be processed and *averaged into a single nucleosome with a very poor density of Rpd3S*. These findings indicate that the engagement of Rpd3S with di-nucleosome is very complicated due to its dynamic and multivalent engagement feature as nicely suggested by this reviewer. However, our current structure did support that a single nucleosome is the primary target for Rpd3S, given the illustrated extensive and multivalent interactions. The discovery of two Eaf3-Rco1 heterodimers revealed a new layer of multivalence centered on full methylation readout of H3K36me3 by Rpd3S (Figure 3a-d). This also explains why a high resolution structure of Rpd3S with a single nucleosome has been achieved.

Regarding the possible mode of di-nucleosome engagement, we performed modelling analysis based on the single nucleosome complex structure. As summarized in Extended Data Figure 20 below, we propose three possible di-nucleosome engagement modes. In mode 1, while the CHD domains of Eaf3-A and Eaf3-B, along with Rco1-A PHD1, recognize one nucleosome, the PHD1 finger of Rco1-B is well-positioned for unmodified histone H3K4 readout of an adjacent nucleosome ~40 bp away (Extended Data Fig. 20a). This may suggest a mechanism by which Rpd3S spreads from one nucleosome to another for efficient deacetylation. Alternatively, in modes 2 and 3, when the stoichiometry of Rpd3S to nucleosome is high, two Rpd3S may simultaneously bind to two adjacent nucleosomes (Extended Data Fig. 20b, c), cooperatively recognizing a di-nucleosome unit with optimal linker DNA length, such as 30-40 bp, for catalysis. Further structural studies are needed to fully understand the recognition mechanism of di-nucleosome engagement by Rpd3S.

Extended Data Figure 20. The catalytic models of Rpd3S with di-nucleosome. a. The two CHD domains of Eaf3-A and Eaf3-B with PHD1 of Rco1-A are involved in recognizing one nucleosome, and PHD1 of Rco1-B is involved in recognizing another nucleosome. **b** and **c.** Two Rpd3S can bind to two nucleosomes, respectively, at a suitable 40bp linker DNA length. The latter Rpd3S complexes may be assembled on two nucleosome discs respectively.

2. How H3 and H4 tails are simultaneously deacetylated on the known *in vivo* template for Rpd3S (a di-nucleosome). The significance of the work would be elevated by addition of using di-nucleosomes in the study to determine how H3 and H4 tails are deacetylated under the *in vivo* conditions that are known for Rpd3S.

Response: Our structural studies and designer nucleosome-based enzymatic assays suggest that H3 and H4 deacetylation are likely dynamically proceeded in two different modes. On one hand, histone H3K36me3 dual mark readout by Rpd3S over a single nucleosome positions the catalytic center of Rpd3 next to the histone H4 N-terminal tail for efficient deacetylation (Fig. 6e). On the other hand, coordinated readout of H3K4un by Rco1-A PHD1 and of H3K36me3 by Eaf3-B CHD over one histone H3 tail determined the deacetylation activity of Rpd3S toward histone H3 (Fig. 6f). Our *in vitro* and *in vivo* mutagenesis (7mu and E260K/D261K) studies support these models (Fig. 5a, 6b). Our current structural study of Rpd3S with H3K36me3 dual mark nucleosome mainly captures the catalytic mode of H4 deacetylation. However, the dynamic engagement feature has been reflected by the existence of “loose” and “close” Rpd3S-Nucl states (Supplementary Video 5). Future studies may be needed to capture other catalytic modes of Rpd3S on either H3 deacetylation or di-nucleosome engagement.

Figure 6. Schematic models of the Rpd3S complex bound to the H3K36me3 modified nucleosome on histone H4 deacetylation (e) and histone H3 deacetylation (f).

3. The inability of H3K9ac is interesting but the authors forgot to consider that Rtt109 acetylates this site (and K56) to promote Asf1 histone exchange. Does this mean that Rpd3S is not involved in histone exchange?

Response: Thanks for this valuable comment. Deleting Set2 in yeast leads to an increase in histone exchange

and the accumulation of H3K56ac over transcribed genes (Venkatesh, S. et al., *Nature* 489, 452-U145, 2012). However, H3K56ac within a nucleosome cannot be erased by Rpd3S due to its buried nature and the need for an intact nucleosome for Rpd3S engagement. This finding is consistent with previous functional studies (Celic, I. et al., *Curr Biol* 16, 1280-1289, 2006) and our yeast knockout experiments (Extended Data Fig. 19). Rtt109, in partnership with histone chaperones Asf1 and Vps75, serves as a writer for H3K56ac and H3K9ac, regulating histone exchange (D'Arcy, S. & Luger, K. *Curr Opin Struct Biol* 21, 728-734, 2011). The inability of Rpd3S to deacetylate H3K56ac and H3K9ac suggests that Rpd3S does not directly downregulate histone exchange by counteracting Rtt109. Set2 methylation of H3K36 likely indirectly suppresses histone exchange by inhibiting cryptic transcription following H3K36me3-mediated H3 and H4 deacetylation by Rpd3S.

Extended Data Figure 19. The H3K56ac modification is not directly affected by the Rpd3S complex *in vivo*. Western blot shows H3 acetylation levels at different sites in Rpd3S wild-type and Rco1-deleted strains. The alterations observed in H3K9ac and H3K56ac are not significant in comparison to other H3 sites in Rco1-deleted strains.

4. Minor comment: the writing and clarity of the text needs some slight improvement.

Response: Corrected! Thanks.

Referee #2(Remarks to the Author):

The Rpd3S complex is a histone deacetylase belonging to the class I HDAC family that is involved in transcriptional repression by deacetylating histones through recognition of H3K36me3. Its three-dimensional structure and the mechanism of histone deacetylation were not elucidated to date. In this study, the authors successfully resolved the three-dimensional structure of the yeast Rpd3S catalytic core complex, a giant complex of 0.6Mda on its own, and the structure of the Rpd3S complex bound to the H3K36me3 nucleosome using cryo-electron microscopy. They also generated structure-based mutants and performed various site-specific deacetylation assays, leading the authors to propose a unique mode of nucleosomal substrate recognition mechanism by the Rpd3S complex. This paper provides the first structural description of the class I HDAC family and a well-dissected mechanistic overview of the recognition of H3K36me3 by the Rpd3S complex and its

binding mode to histone H3 and H4 tails, findings that can be considered as very important for understanding the epigenetic regulation of chromatin by the class I HDAC family. There is no doubt that this study will contribute to the development of research on the Rpd3S complex. The quality of the cryo-EM structure determined, as well as the *in vitro* and *in vivo* analyses are satisfactory. The manuscript deserves to be published, provided that the authors address the comments listed below.

Response: We are grateful for the reviewer's endorsement and favorable comments!

Major comments:

1. The authors should include the H3K36me0 control in the HDAC assay of Rpd3S Rco1 E260K/D261K and Rco1 7mu, mutants impaired in their deacetylation activities, because we do not know whether the results obtained occur in an H3K36me3-dependent manner.

Response: We appreciate the comments provided by this reviewer. In order to investigate the difference between H3K36me0 and H3K36me3 in terms of deacetylation activities, we synthesized combinatorially modified histones H4(K5acK8acK12acK16ac), H3(K9acK14acK18acK23acK27acK36me0), and H3(K9acK14acK18acK23acK27acK36me3), and reconstituted a designer nucleosome for enzymatic assays. Our results indicate that the presence of H3K36me3 significantly increases the level of deacetylation by Rpd3S, which is consistent across both WT and mutant samples. These new findings have been incorporated into Extended Data Fig. 15.

Extended Data Figure 15. H3K36me3 promotes the catalytic activity of Rpd3S on nucleosome substrates.

A representative HDAC assay measuring the activity of Rpd3S complex containing wild-type (a), mutants of PHD1 (b), and left arm region (c) on H3K36me3 and H3K36me0 nucleosomes. The reaction products were identified using Western blot.

2. The authors analyzed the effects of *Rco1* E260K/D261K and *Rco1* 7mu *in vitro* by HDAC assay, but only *Rco1* E260K/D261K was analyzed *in vivo*. Therefore, the authors should investigate the effects of *Rco1* 7mu *in vivo* as well.

Response: Thank you for your suggestion! We have analyzed the differences in H3 and H4 acetylation levels *in vivo* between the wild type strain and the *Rco1* 7mu mutant. Our findings are in line with the results of our *in vitro* enzymatic assays. These new results have been included in the new Fig.6a, b.

Figure 6. *In vivo* modification crosstalk studies. a. The test cryptic transcription phenotype caused by *Rco1* E260K/D261K or 7mu mutants in a STE11-HIS reporter strain (YBL853). **b.** Western blot shows H3 and H4 acetylation levels at different sites in wild-type, E260K/D261K, and 7mu mutants of *Rco1*.

3. To investigate the importance of H3K9ac, the authors generated an H3K9R mutant and studied its effect *in vivo*. The authors state that the introduction of the H3K9R mutation reduced acetylation of histones H3 and H4 by HATs such as NuA3 and NuA4. However, it is also possible that it was enhanced deacetylation by Rpd3S. This could be easily verified by the authors by performing an HDAC assay with H3K9R *in vitro*.

Response: Thank you for your valuable comment. We conducted enzymatic assays using a reconstituted designer nucleosome containing combinatorially modified histones H3(K9RK14acK18acK23acK27acK36me3) and H4(K5acK8acK12acK16ac) to investigate the effect of H3K9R mutation on the deacetylation activities of Rpd3S towards histone H3 and H4. We found that the change of H3K9R did not enhance the deacetylation activities of Rpd3S towards histone H3 and H4, but instead reduced its deacetylation activities to some extent (Extended Data Fig.15a and Extended Data Fig.16).

This suggests that the reduction of acetylation levels of histone H3 and H4 *in vivo* is not caused by the promotion of Rpd3S activity by H3K9R mutation.

Extended Data Figure 15a

Extended Data Figure 16

Extended Data Figure 15. H3K36me3 promotes the catalytic activity of Rpd3S on nucleosome substrates. a. A representative HDAC assay measuring activity of Rpd3S complex on H3K36me3 nucleosome.

Extended Data Figure 16. The catalytic activity of Rpd3S on H3K9RK36me3 nucleosome. A representative HDAC assay measuring activity of Rpd3S complex on H3K9RK36me3 nucleosome.

4. In the introduction, the authors mention an interesting study showing that deacetylase activity may vary with the length of the linker DNA. However, the authors do not discuss this at all in the discussion. It would be very interesting if the authors could talk over this hypothesis based on their structure obtained in this study, to further the discussion.

Response: Thanks for the suggestion! We have discussed it and incorporated the necessary updates in the discussion in the manuscript (lines 328-339).

5. In Figure 4, the angles of the overall structure and those of the close-up view are very different and are difficult to understand. The authors should correct the angles to make the figure more understandable to readers.

Response: Thank you for bringing this to our attention. We have made corresponding corrections to ensure that the orientation of the overall structure and the close-up view in Figure 4 are consistent, thus improving their clarity.

Figure 4. Interface details between Rpd3S complex and H3K36me3 nucleosome. **a.** Model of core Rpd3S complex bound to the H3K36me3 modified nucleosome on histone H4 deacetylation. The positions of interaction are marked with dotted line and numbers. **b.** Detailed view of the location between H4 N-terminus and Rpd3 in close state shown in dotted line. **c.** Detailed view of the interactions between CHD and the H3K36me3 modified nucleosome. The residues of CHD and the nucleotides of nucleosomal DNA involved in recognition and H3 tail residues are shown as sticks. Selected hydrogen bonds are shown as red dashed lines. **d.** Detailed view of interactions between Rco1-MID and linker DNA of nucleosome. Residues at the interface are depicted as sticks. **e.** Detailed view of interactions between Sin3 and nucleosomal DNA at SHL+2.5. The residues of Sin3 and the nucleotides of nucleosomal DNA involved in recognition are shown as sticks. Selected hydrogen bonds are shown as red dashed lines.

6. Extended Data Figure 12: it is not clear whether it is Eaf3-A CHD at SHL+7 or Eaf3-B CHD at SHL+1 that binds to H3K36me3. Also, the binding positions of the nucleosomes are different, so the authors should

show independently the two nucleosomes with the different binding positions in the figure.

Response: Thank you for bringing this to our attention. We have made the necessary corrections to Extended Data Figure 12 to clarify which CHD domain of Eaf3 binds to H3K36me3 at which nucleosome position. Additionally, we have included separate illustrations of the two nucleosomes with their distinct binding positions to better convey this information. We appreciate your feedback and hope that these changes improve the clarity of our research.

Extended Data Figure 12. Cryo-EM densities of CHD domain and histone H3K36me3. **a.** Detailed view of interactions between CHD and H3 tail. Close-up views of the loop of CHD and H3 tail for interactions with cryo-EM densities shown as meshes. **b.** Detailed view of the interactions between Eaf3^A-CHD and the H3K36me3 modified nucleosome. **c.** Detailed view of the interactions between Eaf3^B-CHD and the H3K36me3 modified nucleosome. The residues of CHDs and H3 tail residues are shown as sticks. The positions of nucleosomal DNA are labeled with SHL numbers.

Minor comments

1. The following should be added to the cryo-EM structural analysis figures:
- Extended Data Figure 2a, 9a: scale bar

- Extended Data Figure 2b, 9b: box size

- Extended Data Figure 2, 9: Euler angle distribution map

2. P.6 line135: "A PAH4 domain (1143-1216) has been reported to exist within Sin3." The authors need to cite a reference.

3. Extended Data Figure 10a: the nucleosome map is inverted. The authors must correct it.

4. A space is required between the number and the unit below.

Figure 5a bottom right: 25min → 25 min

Figure 6C: 100µg/ml → 100 µg/ml

Response: Thanks, all corrected!

Referee #3(Remarks to the Author):

The manuscript by Guan and colleagues describes the structure of the yeast Rpd3S histone deacetylase complex in its apo and nucleosome bound forms. Rpd3S is an important histone deacetylase complex that functions in gene regulation and suppression of aberrant cryptic transcription by interacting with methylated H3K36 that is associated with transcribed genes. This work describes the structure of one of the most well-known and studied epigenetic erasers in chromatin biology, and reveals the molecular basis of recognition between these enzymatic complexes and their nucleosome substrates, and how that recognition allows tuning of the catalytic targets of the deacetylase enzyme which is explored further using structure-guided biochemical assays. HDAC-nucleosome structures are beginning to emerge in the field, but this work provides one of the most detailed and comprehensive studies to date. The work is high quality and a major advance in chromatin biology. I imagine it will be of great interest to the community and beyond, and thus suitable for publication in Nature. I only have these minor comments:

Response: We appreciate the reviewer's support and favorable comments!

1. A knotted coil is a very rare topology. How sure are the authors that this is really so, given that the supplementary figure 4 shows a significant un-modelled portion of the coil, but whose path is threaded through a loop (and forming the knot) in their schematic?

Response: This is an interesting question. Knotted proteins have been identified in all kingdoms of life. High conservation of knotted motifs and their location (usually) in enzymatic active sites indicates that knots are crucial for protein function, such as an enhanced kinetic stability. At first, we were uncertain about the model construction of this area. While we thought it might be a unique knotted coil, it was difficult to confirm based solely on the map. However, AlphaFold's prediction of the structure aligned perfectly with the assumption of a knotted coil, which was a pleasant surprise and gave us a high level of confidence. We have included this prediction as new Extended Data Figure 4d in the revised manuscript, and we consider it to be a reliable one.

Extended Data Figure 4. Detailed structure of Rco1-SBD. **a.** Stereo view of the SBD domain of Rco1-A shown in cartoon form. Detailed structures of visible SBD domain of Rco1-A (**b**) and Rco1-B (**c**). Residues are depicted as sticks. **d.** Stereo view of the SBD domain of Rco1-A predicted by AlphaFold.

2. I found the paragraph describing the two arms (lines 102-119) extremely difficult to interpret. It took a lot of effort and i'm still unsure whether i understand the authors description. One of the phrases that were the most confusing are "...the helix element of Rco1-B SBD participates in SBD surface engagement despite the invisibility of the knotted coil motif". Admittedly, the structure is very complex but i'd welcome a clearer description if possible.

Response: Thank you. We have renamed the SBD regions in Figure 2a and 2b (SBD-R and SBD-L) to improve clarity. The updated article now states that "On the left arm, MRGEaf3-B and MIDRco1-B form an SBD-L surface that interacts with Rco1-A SBD and subsequently Sin3 PAH3 for assembly (Fig. 2a). By contrast, the "MRG-PHD1-MID" module in the right arm uses the opposite SBD-R surface that involves MRGEaf3-A and PHD1Rco1-A for Rpd3S assembly (Fig. 2a). Surprisingly, the helix element of Rco1-B C-terminal SBD domain also participates in SBD-L surface engagement despite the invisibility of the knotted coil motif." These changes have been incorporated into new Figure 2a and b.

Figure 2. Interaction and regulation between subunits. a. A global view of the Rpd3S complex highlighting the heterodimers of Rco1-Eaf3. The PHD1 (light brown), MID (reseda) domain of Rco1 and MRG (blueviolet) domain of Eaf3 are shown in surface representation, while others are shown in cartoon form. The dotted lines represent the SBD surfaces (SBD-R and SBD-L) at different angles. **b.** Close-up view of interactions of SBD domains of Rco1 and PAH3 domain of Sin3 in SBD-L surface and bridge region.

3. The structural basis described for crosstalk between unmodified H3K4 and H3 deacetylation is reasonable, and supported by some studies (e.g. reference 13). But it would be far more compelling if the deacetylation experiment performed in Figure 5a was applied to nucleosomes bearing H3K4me1/2/3. If their model is correct, this should reduce/prevent deacetylation in the same manner as the Rco1 mutants.

Response: Thank you for your valuable input. Based on research conducted by the Brian D. Strahl group (reference 19), it has been found that the trimethylation of K4 (H3K4me3) reduces PHD1's ability to bind to the N-terminal H3₁₋₂₀ peptide. We synthesized modified histones H3(K4me3K9acK14acK18acK23acK27acK36me3), H3(K4me0K9acK14acK18acK23acK27acK36me3), and H4(K5acK8acK12acK16ac) and reconstituted designer nucleosome samples for enzymatic assays. Our experiments revealed that the presence of H3K4me3, like the Rco1 E260K/D261K mutant, affects PHD1's binding to H3 N-terminal and significantly disrupts Rpd3S's deacetylation activity towards histone H3 but not H4 to some extent. These results have been incorporated in Extended Data Fig.14 and Extended Data Fig.15a.

Extended Data Figure 15a

Extended Data Figure 14

Extended Data Figure 15. H3K36me3 promotes the catalytic activity of Rpd3S on nucleosome substrates. a. A representative HDAC assay measuring activity of Rpd3S complex on H3K36me3 nucleosome.

Extended Data Figure 14. The catalytic activity of Rpd3S on H3K4me3K36me3 nucleosome. A representative HDAC assay measuring activity of Rpd3S complex on H3K4me3K36me3 nucleosome. The reaction products were identified using Western blot.

Reviewer Reports on the First Revision:

Referees' comments:

Referee #1 (Remarks to the Author):

The authors have adequately addressed my concerns. I appreciate the inclusion of the di-nucleosomes models that highlight possible mechanisms of Rpd3S binding and spreading. I agree future studies will be needed to determine how the more complex mechanism of deactivation works. Nonetheless, the work is of high significance and provides a new mechanistic basis for how H3 and H4 tails become deacetylated that will be of broad interest.

Referee #2 (Remarks to the Author):

In this study, the authors have succeeded in resolving the three-dimensional structure of the Rpd3S catalytic complex and have provided a well-dissected mechanistic overview of Rpd63 and its nucleosomal engagement. There is no doubt that this paper will contribute to further research on the Rpd3S complex.

In the revised manuscript, the authors have properly addressed both major and minor comments. The quality of the resulting manuscript is very satisfactory; therefore, I believe that this study deserves to be published in Nature.

Referee #3 (Remarks to the Author):

I thank the authors for their responses and efforts in editing the manuscript and the additional experiments performed. The manuscript is much improved and I approve its publication.